# Receptor-based mechanism of relative sensing and cell memory in mammalian signaling networks

**Eugenia Lyashenko[1†‡], Mario Niepel[2†§], Purushottam D Dixit[1,3†#], Sang Kyun Lim[2], Peter K Sorger[1,2], Dennis Vitkup[1,4,5]\***

[1]Department of Systems Biology, Columbia University, New York, United States; [2]HMS LINCS Center Laboratory of Systems Pharmacology, Department of Systems Biology, Harvard Medical School, Boston, United States; [3]Department of Physics, University of Florida, Gainesville, United States; [4]Center for Computational Biology and Bioinformatics, Columbia University, New York, United States; [5]Department of Biomedical Informatics, Columbia University, New York, United States

**Abstract** Detecting relative rather than absolute changes in extracellular signals enables cells to make decisions in constantly fluctuating environments. It is currently not well understood how mammalian signaling networks store the memories of past stimuli and subsequently use them to compute relative signals, that is perform fold change detection. Using the growth factor-activated PI3K-Akt signaling pathway, we develop here computational and analytical models, and experimentally validate a novel non-transcriptional mechanism of relative sensing in mammalian cells. This mechanism relies on a new form of cellular memory, where cells effectively encode past stimulation levels in the abundance of cognate receptors on the cell surface. The surface receptor abundance is regulated by background signal-dependent receptor endocytosis and down-regulation. We show the robustness and specificity of relative sensing for two physiologically important ligands, epidermal growth factor (EGF) and hepatocyte growth factor (HGF), and across wide ranges of background stimuli. Our results suggest that similar mechanisms of cell memory and fold change detection may be important in diverse signaling cascades and multiple biological contexts.

**\*For correspondence:**
dv2121@cumc.columbia.edu

[†]These authors contributed equally to this work

**Present address:** [‡]Human Target Validation Core, Biogen, Cambridge, United States; [§]Ribon Therapeutics, Cambridge, United States; [#]Kanaph Therapeutics, Seuol, South Korea

**Competing interests:** The authors declare that no competing interests exist.

## Introduction

In biological systems, concentrations of extracellular signaling molecules, such as hormones and growth factors, often vary by orders of magnitude. Therefore, the ability to sense relative rather than absolute signals, that is detect fold changes in extracellular cues, is critical for making accurate decisions in different biological contexts (*Alon, 2019*). Relative sensing requires both the ability to store memories of past environmental stimuli and the capacity to quickly and efficiently compute relative signals (*Adler and Alon, 2018*).

Relative sensing of environmental inputs has been previosuly investigated in bacteria, with the *E. coli* chemotaxis being a classic example (*Mesibov et al., 1973*; *Barkai and Leibler, 1997*; *Alon et al., 1999*; *Shoval et al., 2010*). Studies have also explored relative sensing in a variety of eukaryotic systems. When responding to constant stimuli, experiments with the signaling proteins ERK (*Cohen-Saidon et al., 2009*) and β-catenin (*Goentoro and Kirschner, 2009*) showed that fold changes in their nuclear activity were robust to cell-to-cell variability (*Cohen-Saidon et al., 2009*) and variability in signaling network parameters (*Goentoro and Kirschner, 2009*). These observations suggested that gene expression of target genes may respond, at the single cell level, to fold changes rather than absolute activities of these proteins. Later studies of the NF-κB

(*Lee et al., 2014*) and TGF-β/SMAD pathways (*Frick et al., 2017*) also showed that genes directly controlled by these proteins often respond to their fold changes at the single cell level. Recent work has explored relative sensing at the organism level in plants, where the chlorophyll activity was found to be proportional to the fold change in external light intensity (*Tendler et al., 2018*).

Despite the insights gained in the aforementioned studies, the molecular mechanisms allowing cells to detect fold changes in extracellular stimuli are not well understood. The key unresolved questions are: (1) where and how the memories of background extracellular stimuli are stored within the cell, (2) what makes these memories specific to particular stimuli, and (3) how the cells subsequently use the stored memories to compute fold changes.

In this work, using the growth factor-activated PI3K/Akt signaling pathway, we describe a novel non-transcriptional mechanism of relative sensing in mammalian cells. The mechanism operates on fast timescales of dozens minutes to hours, and across more than an order of magnitude of extracellular background stimuli. We derive key aggregate parameters of the signaling cascade that determine the accuracy and the background range of relative sensing. We also experimentally validate the accuracy of relative sensing by stimulating cells with multiple fold changes of two physiologically important ligands, EGF and HGF. Furthermore, we demonstrate that ligand relative sensing is reliably propagated to an important downstream target of the PI3K/Akt pathway.

## Results

Stimulation of mammalian cells with growth factors elicits a variety of context-dependent, phenotypic responses, including cell migration, proliferation, and cell survival (*Cantley et al., 2014*). Akt serves as a central hub of multiple growth factor-activated signaling cascades (*Hemmings and Restuccia, 2012*). Naturally, Akt phosphorylation-dependent (pAkt) pathways are implicated in multiple human diseases, such as many types of cancers (*Engelman, 2009*; *Hemmings and Restuccia, 2012*), diabetes (*Whiteman et al., 2002*), and psychiatric disorders (*Gilman et al., 2012*; *McGuire et al., 2014*).

To understand how the immediate-early dynamics of the Akt pathway depend on the background level of growth factors, we used immunofluorescence to quantify the levels of pAkt in epidermal growth factor (EGF)- stimulated human non-transformed mammary epithelial MCF10A cells (Materials and methods, *Figure 1—figure supplement 1*). Within minutes of continuous stimulation with EGF pAkt reached maximum response, and then decayed to low steady state levels within hours (*Figure 1a*). The resulting steady state pAkt levels were approximately independent of the EGF stimulus, indicating an approximately adaptive response (*Friedlander and Brenner, 2009*; *Shoval et al., 2010*; *Figure 1—figure supplement 2*). In the sensitive range of EGF concentrations, maximal pAkt response was approximately proportional to the logarithm of the EGF stimulus (*Figure 1b*). Quantitative western blot experiments demonstrated that in this logarithmic regime, pAkt levels were approximately linearly proportional to the phosphorylation level of EGF receptors (EGFRs) (*Figure 1—figure supplement 3*). The logarithmic dependence of EGFR phosphorylation levels on EGF stimulation has been previously attributed to a mixture of receptor species with varying affinities to the ligand, negative cooperativity of ligand binding to receptor dimers, and oligomeric aggregation of receptors (*Kawamoto et al., 1983*; *Chatelier et al., 1986*; *Wofsy et al., 1992*; *Macdonald and Pike, 2008*; *Huang et al., 2016*).

Continuous stimulation with EGF resulted in the abundance of cell-surface EGF receptors (sEGFR) also decreasing proportionally to the logarithm of the background EGF level, and reaching a new steady state within hours (*Figure 1c*). Notably, prior exposure with EGF desensitized cells to subsequent EGF stimulations in a quantitative manner. When we first pre-exposed cells to different levels of EGF for 3 hr and then stimulated them with the same final EGF concentration (2 ng/ml)), the maximal pAkt response decreased monotonically with increasing pre-exposure EGF levels (*Figure 1d*). These experiments demonstrate that the pAkt response to an abrupt EGF stimulation is strongly affected by background EGF levels, and that this effect is likely mediated by the endocytosis-based removal of activated EGFRs from the cell surface (*Wiley et al., 1991*).

Using pharmacological perturbations of the EGFR/Akt pathway, we confirmed that the desensitization of the phosphorylation response (*Figure 1d*) was likely due to receptor-based mechanisms upstream of Akt activation, and did not depend on its downregulation, for example, through phosphorylation-dependent Akt degradation (*Wu et al., 2011*). Specifically, we used SC79, a small

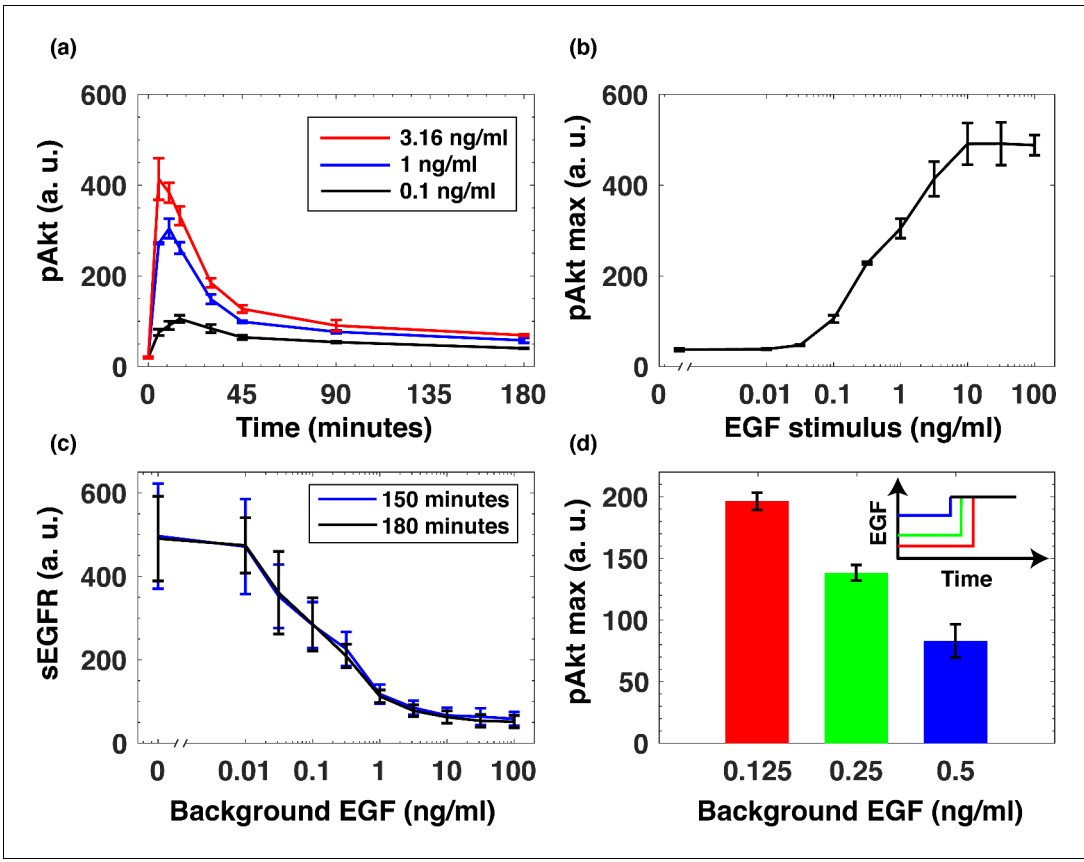

**Figure 1.** EGF-induced Akt phosphorylation and desensitization in MCF10A cells. (**a**) Temporal profiles of phosphorylated Akt (pAkt) in cells exposed to increasing stimulation with extracellular EGF (see inset). (**b**) Maximal pAkt response as a function of EGF stimulation. (**c**) Steady state levels of surface EGFR (sEGFR) after 150 and 180 min of stimulation with a constant level of EGF. (**d**) Desensitization of the maximal pAkt response to an abrupt EGF stimulation. MCF10A cells were pre-treated with increasing background doses of EGF (x-axis) for three hours, followed by a second abrupt stimulation with the same concentration of EGF (2 ng/ml); the inset shows a schematic illustration of the experimental protocol. In all subpanels, error bars represent the standard deviation of n = 3 technical replicates. Source data: pakt_timecourses_first_step.mat and segfr_150_180mins.doseresponse. mat (available in *Source code 1*).

The online version of this article includes the following figure supplement(s) for figure 1:

**Figure supplement 1.** Representative immunofluorescence stains of Akt phosphorylation and cell surface EGF receptor levels.

**Figure supplement 2.** Adaptation of steady state pAkt response.

**Figure supplement 3.** Akt phosphorylation response is linearly related to EGFR phosphorylation response.

**Figure supplement 4.** pAkt activation using pharmacological intervention.

---

molecule which promotes Akt phosphorylation even in the absence of extracellular ligands (*Jo et al., 2012*). Unlike the desensitization observed in the growth factor-induced pAkt response (*Figure 1d*), the pAkt response following stimulation with SC79 did not depend on the background EGF pre-exposure (*Figure 1—figure supplement 4*). This result supports the conclusion that the EGF desensitization mechanism was upstream of Akt.

To understand how background EGF levels affect the pAkt response to subsequent EGF stimulation we next constructed an ordinary differential equation (ODE) model of EGF-dependent Akt phosphorylation. The model included several well-established features of the EGFR signaling cascade (*Chen et al., 2009*), such as endocytosis and degradation of activated receptors (Materials and methods) (*Figure 2a*). We constrained the ranges of model parameters based on literature-derived estimates (*Supplementary file 1a*), and fitted the model using experimental data on pAkt time courses (*Figure 1a*) and steady state sEGFR levels (*Figure 1c*) at different doses of EGF

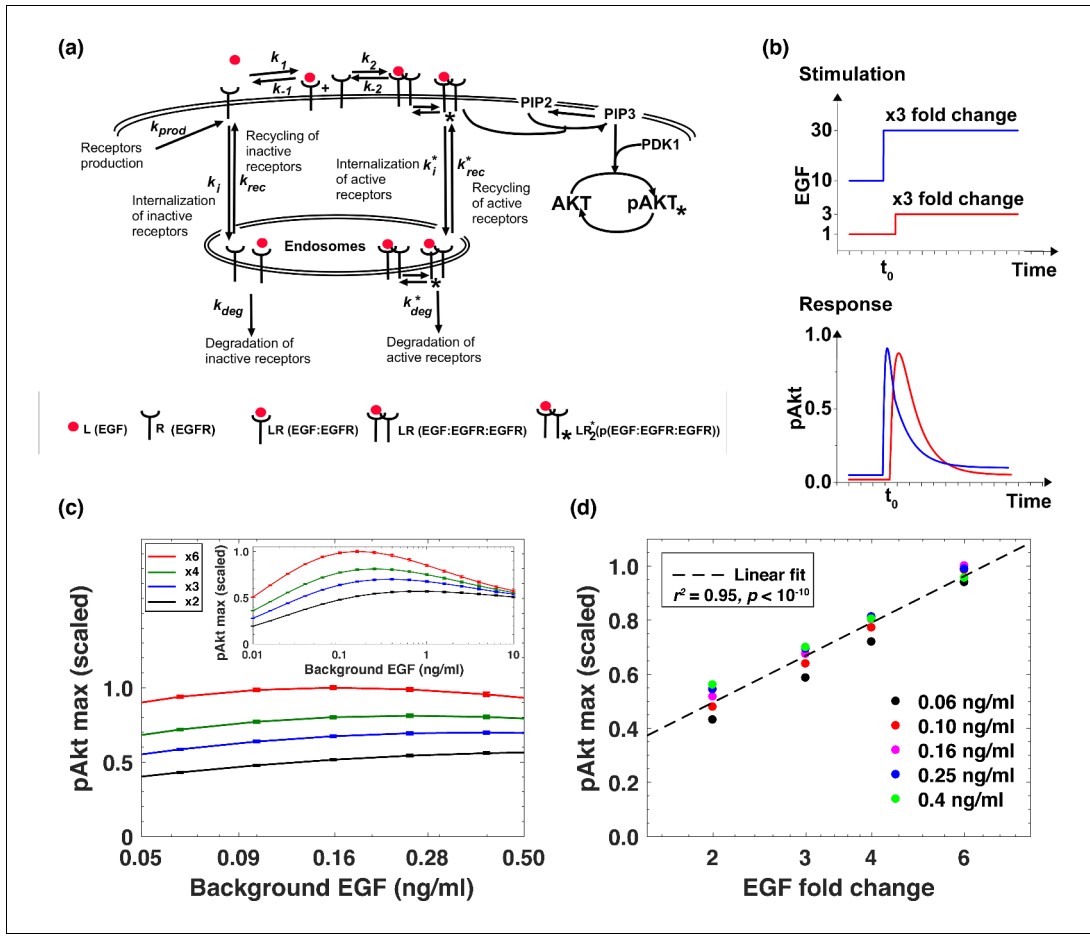

**Figure 2.** Computational model demonstrates the ability of the system to sense relative changes of EGF levels. (**a**) Schematic of the computational model of the EGFR signaling cascade leading to phosphorylation of Akt. Rate constants marked with asterisks correspond to reactions associated with activated (phosphorylated) receptors. Only a subset of reactions in the network are shown for brevity. (**b**) In silico protocol used to explore relative sensing, showing the temporal profiles of EGF stimulation (top) and the corresponding profiles pAkt response (bottom). Cells were first exposed to various background EGF stimulations (blue and red) and were next subjected to the same abrupt fold change in EGF at time $t_0$. The resulting maximal pAkt responses were similar for the same EGF fold change independent of background EGF stimulation, indicating relative sensing. (**c**) The maximal pAkt response observed after exposing the ODE model in silico to different background EGF levels (x axis), followed by a 2-, 3-, 4-, or 6- fold increase (different colors) in EGF; inset shows pAkt response over a wider range of background EGF levels. (**d**) Maximal pAkt responses (y axis) induced by stimulation with different EGF background levels (data points with the same shape and color) were combined and plotted as a function of the EGF fold change (x axis). Dashed line represents log-linear fit to data (Pearson's $r^2$ = 0.96, regression $p$ value < $10^{-15}$). In all subpanels, error bars represent the standard deviation of n = 10 model fits. Source code: https://github.com/dixitpd/FoldChange/.

The online version of this article includes the following figure supplement(s) for figure 2:

**Figure supplement 1.** Dynamical model fits to experimental data.

stimulations. We then used simulated annealing to optimize model parameters (Materials and methods, *Figure 2—figure supplement 1*), and considered multiple distinct parameter sets from the optimization runs for further computational analysis.

Using the fitted dynamical model (*Figure 2—figure supplement 1*), we explored the ability of the Akt pathway to respond to relative, rather than absolute, changes in EGF levels. To that end, we simulated the pAkt response by exposing the model *in silico* to a range of background EGF levels followed by different abrupt fold change increases in EGF concentration (*Figure 2b*). The model predicted that the maximal pAkt response indeed depends primarily on the EGF fold change relative to

the background stimulation levels (*Figure 2c*). This relative sensing of EGF stimuli occurred over an order of magnitude of background EGF concentrations, and the resulting pAkt response was approximately proportional to the logarithm of the EGF fold change (*Figure 2d*). Notably, the model predicted relative sensing exactly in the range of EGF background concentrations where sEGFR endocytosis was sensitive to background ligand stimulation. At low EGF background concentrations (<0.01 ng/ml), no substantial sEGFR removal was predicted at the steady state (*Figure 2—figure supplement 1*), and consequently there was no significant desensitization of the pAkt response. In that regime, the pAkt response to an abrupt fold change depended primarily on the absolute EGF level. In contrast, at high background EGF concentrations (>1 ng/ml), a large fraction of sEGFR was already removed from cell surface and consequently the network responded only weakly to further EGF stimulation.

Next, we experimentally tested the model-predicted relative sensing in MCF10A cells. Cells were first treated with various background EGF concentrations for three hours to ensure that sEGFR reached steady state levels (*Figure 1c*), and that pAkt had decayed after a transient increase (*Figure 1a*). As in the computational analysis (*Figure 2b*), cells were then exposed to different fold changes in EGF levels; pAkt levels were measured at 2.5, 5, 10, 15, 30 and 45 min after the step increase in EGF stimulation (*Figure 3—figure supplement 1*). Similar results were observed in two independent biological replicates (*Figure 3—figure supplements 1*, *2* and *3*), and the experiments confirmed the predictions of the computational model that maximal pAkt response depends primarily on the fold change of EGF and not its absolute concentration (*Figure 3a*, *Figure 3—figure supplement 4*). Specifically, across more than an order of magnitude of EGF background concentrations (0.03–0.5 ng/ml) the same EGF fold change (lines with the same colors in *Figure 3a*) elicited similar

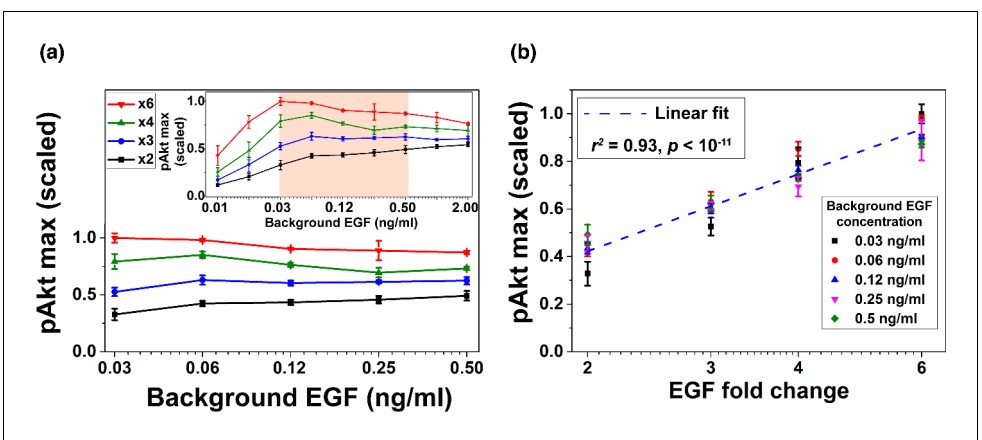

**Figure 3.** Experimental validation of EGF relative sensing by pAkt in MCF10A cells. (a) The maximal pAkt responses after exposing cells to different background EGF levels (x axis) for 3 hr, followed by 2-, 3-, 4-, and 6-fold increases (different colors) in EGF. Inset shows experimental pAkt response over a wider range of background EGF levels. (b) Maximal pAkt responses (y axis) to fold changes in EGF depended approximately logarithmically on the fold change. Maximal pAkt responses induced by stimulation with various EGF background levels (data points with the same shape and color) were combined and plotted as a function of the EGF fold change (x axis). Dashed line represents log-linear fit to the data (Pearson's $r^2 = 0.93$, regression $p$ value < $10^{-11}$). In all subpanels, error bars represent the standard deviation of n = 3 technical replicates. Source data: expt_data.mat (available in *Source code 1*).

The online version of this article includes the following figure supplement(s) for figure 3:

**Figure supplement 1.** Dynamics of pAkt responses to step increases in EGF.

**Figure supplement 2.** The scatter plot of pAkt levels observed in the two biological replicates shown in *Figure 3—figure supplement 1*.

**Figure supplement 3.** Biological replicate of the experiment demonstrating relative sensing of EGF by pAkt (main text *Figure 3*).

**Figure supplement 4.** Maximal Akt phosphorylation response does not depend on the absolute EGF stimulus.

**Figure supplement 5.** Time-integral of pAkt response exhibits relative sensing of EGF.

**Figure supplement 6.** Dynamic time course of pAkt response exhibits relative sensing of EGF fold change.

pAkt responses. The concentration range in which we obsered relative sensing was consistent with recent estimations of in vivo EGF levels (*Pinilla-Macua et al., 2017*). In close agreement with the computational model predictions, the maximal pAkt response was approximately proportional to the logarithm of EGF fold change (*Figure 3b*). Interestingly, in addition to the maximal pAkt response, approximate relative sensing was also observed for the time integral of pAkt levels (*Figure 3—figure supplement 5*), and for the entire time course of pAkt dynamics (*Figure 3—figure supplement 6*).

To better understand the mechanism responsible for the observed relative sensing of extracellular EGF concentration, we next constructed a simplified analytical model of the signaling network (see Appendix). This model revealed that, across a broad range of background concentrations, the steady-state abundance of cell surface receptors $[R]_T$ decreases approximately log-linearly as a function of the background ligand (EGF) concentration $[L]_0$ (*Equation 1* and *Figure 4a*):

$$[R]_T \sim constant - a * \log[L]_0 \tag{1}$$

and that the maximal receptor phosphorylation response $[LR_2^*]_{max}$ depends approximately log-linearly on the level of the subsequent stimulation $[L]_1$ and linearly on the steady-state receptor abundance $[R]_T$ (*Equation 2*, *Figure 4b*):

$$[LR_2^*]_{max} \sim b * \left( \log[L]_1 + \frac{[R]_T}{a} \right) + constant \tag{2}$$

where $a$ and $b$ are numerical constants (Appendix). As a result of these relationships, the phosphorylation response $[LR_2^*]_{max}$ after an increase of ligand concentration from $[L]_0$ to $[L]_1$ depends, in agreement with computational and experimental analyses, approximately on the logarithm of the stimulation fold change $\frac{[L]_1}{[L]_0}$:

$$[LR_2^*]_{max} \sim b \times \left( \log[L]_1 + \frac{[R]_T}{a} \right) \sim b \times \left( \log[L]_1 - \log[L]_0 \right) \sim b \log \frac{[L]_1}{[L]_0} \tag{3}$$

The analytical model (Appendix) also revealed that the range of the background ligand concentrations where relative sensing is observed is primarily determined by two aggregate systems parameters, which we denote $\alpha$ and $\beta$ (*Equations 4 and 5*). The parameter $\alpha$ quantifies the

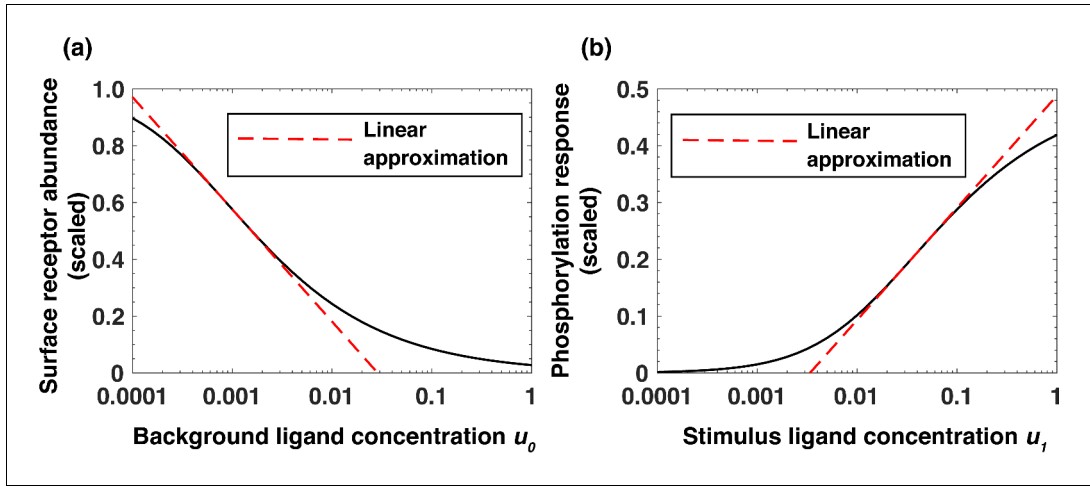

**Figure 4.** Analytical model of the system predicts log-linear relationships leading to receptor-based memory and relative sensing. (a) Approximate log-linear dependence of the scaled steady-state surface receptor abundance $[R]_T$ on the normalized background ligand concentration $u_0 = [L]_0/K_{d1}$, where $K_{d1}$ is the equilibrium dissociation constant of EGF binding to EGFR. (b) Approximate log-linear dependence of the maximal phosphorylation response on the normalized ligand stimulus $u_1 = [L]_1/K_{d1}$. Dashed red lines represent the exact log-linear approximation.

ability of the signaling network to capture the input signal (EGF) and elicit a downstream phosphorylation response. The parameter $\beta$ quantifies the ability of the network to preferentially internalize and degrade active (phosphorylated) receptors relative to inactive (non-phosphorylated) receptors. The two aggregate parameters are expressed as follows:

$$\alpha = \frac{k_p + k_{dp}}{k_{dp}} \times \frac{k_2}{k_{-2}} \times R_0 \tag{4}$$

where $k_p$ is the rate of receptor phosphorylation and $k_{dp}$ is the rate of receptor de-phosphorylation, $k_2$ is the rate of receptor dimerization, $k_{-2}$ is the dissociation rate of receptor dimers, and $R_0$ is the total number of cell-surface receptors at the steady state in the absence of extracellular stimuli and

$$\beta = \frac{k_p}{k_{dp} + k_p} \times \frac{k_i^*}{k_i} \times \frac{\frac{k_{deg}^*}{k_{deg}^* + k_{rec}^*}}{\frac{k_{deg}}{k_{deg} + k_{rec}}} \tag{5}$$

where $k_i^*$, $k_{rec}^*$, $k_{deg}^*$ and $k_i$, $k_{rec}$, $k_{deg}$ are correspondingly the rates of internalization, recycling, and degradation of the active (phosphorylated) and non-active receptors. Notably, an increase in the value of $\alpha$ increases signal sensitivity and receptor dimerization and phosphorylation. This shifts the relative sensing range to lower ligand concentrations (*Figure 5a*). In turn, an increase in the value of $\beta$ increases the fraction of active receptors being internalized and degraded. This increases the range of background ligand concentrations where the relative sensing is observed (*Figure 5b*). Based on the best-fit ODE model parameter sets, we estimate $\alpha \sim 15$ and $\beta \sim 40$ (Appendix). As an example, in *Figure 5* we show the scaled phosphorylation response to a six-fold change in EGF concentration as a function of the scaled background ligand concentration $u_0$ for different values of $\alpha$ (*Figure 5a*) and $\beta$ (*Figure 5b*); the green arrows in the figure represent the predicted range of fold change detection. The model analysis showed that the relative sensing occurs across over an order of magnitude of background ligand concentrations (Appendix). Furthermore, the analytical model revealed that relative sensing does not require receptor dimerization, and similar sensing mechanisms can operate in pathways where signaling is initiated by monomeric receptors (Appendix).

In addition to EGF, Akt phosphorylation can be induced by multiple other ligands, including hepatocyte growth factor (HGF) (*Stuart et al., 2000*) which binds to its cognate receptor cMet

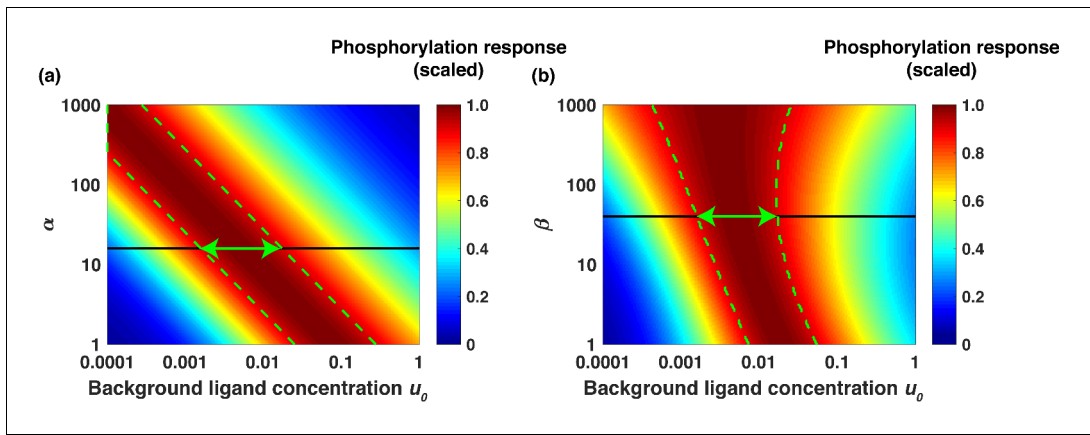

**Figure 5.** Analytical model predicts the range of approximate EGF relative sensing. Scaled phosphorylation response to a six-fold change in extracellular EGF concentration as a function of the scaled background ligand concentration ($u_0 = [L]_0/K_{d1}$). (a) The phosphorylation response as a function of background ligand concentration (x-axis) is shown for different values of the parameter $\alpha$ (y-axis), when the parameter $\beta$ is fixed at $\beta$=40. (b) The phosphorylation response as a function of background ligand concentration (x-axis) is shown for different values of $\beta$ (y-axis), when $\alpha$ is fixed at $\alpha$=15. The colors represent the scaled phosphorylation response. The green dashed lines delineate the parameter ranges where the fold change detection is >90% accurate. The horizontal black lines correspond to the parameter values $\alpha$=15 and $\beta$=40, which were estimated from experimental data fits; the horizontal green double arrows represent the predicted range of relative sensing for the investigated PI3K-Akt cascade.

(*Viticchiè and Muller, 2015*). Similar to EGFRs, upon ligand binding, cMet receptors dimerize (*Kong-Beltran et al., 2004*) and cross-phosphorylate each other; this leads to phosphorylation of multiple downstream targets, including Akt. To investigate the specificity of the receptor-based cell memory to past ligand exposures, we used the two ligands, EGF and HGF, which share many signaling components downstream of their cognate receptors (*Xu and Huang, 2010*). We exposed cells to background doses of either HGF or EGF for three hours, and then stimulated cells using either the same or the other growth factor to elicit pAkt response (*Figure 6a,b*). Pre-exposure with HGF did not substantially downregulate EGF-induced pAkt responses, but substantially

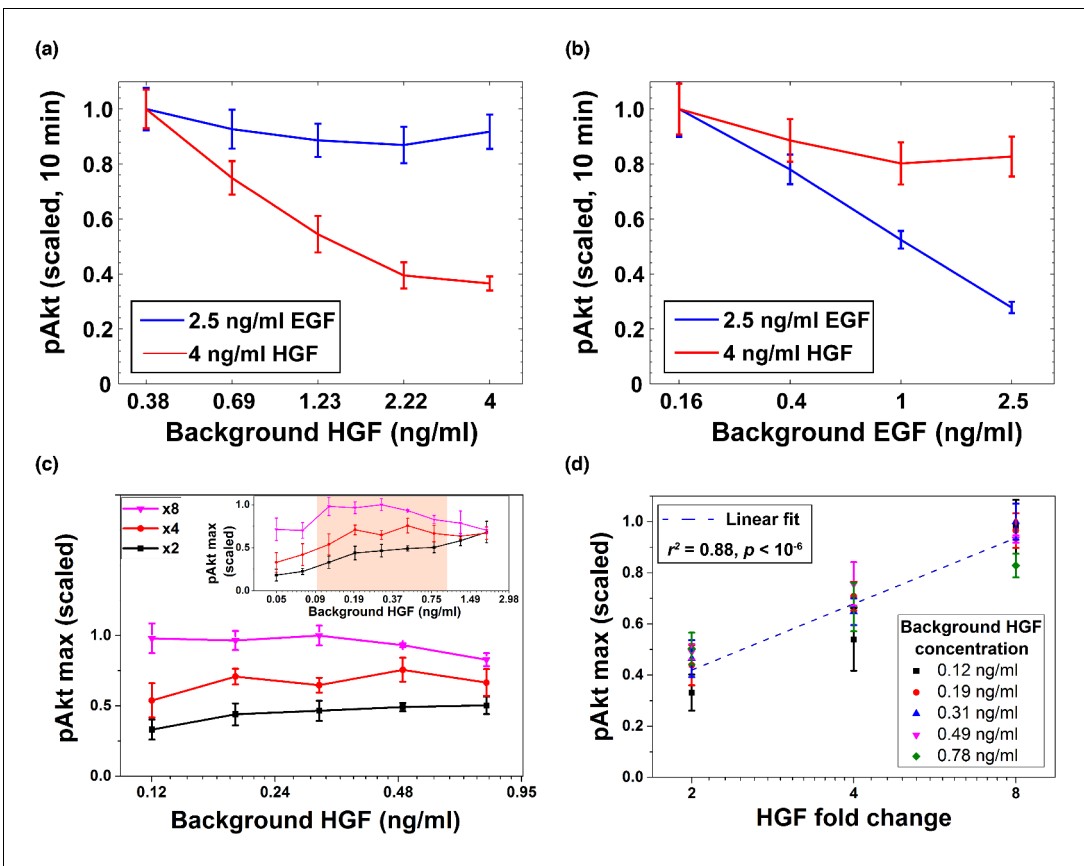

**Figure 6.** Desensitization and ligand-specific cell memory for EGF- and HGF-induced pAkt responses. MCF10A cells were first exposed to various background concentrations of either HGF or EGF for three hours, and then abruptly stimulated using either the same or the other growth factor. pAkt levels were then measured 10 min after the addition of the second stimulus. (a) EGF- (blue, 2.5 ng/ml) or HGF- (red, 4 ng/ml) induced pAkt response in cells pre-exposed with various background concentrations of HGF (x axis). (b) EGF- (blue, 2.5 ng/ml) or HGF- (red, 4 ng/ml) induced pAkt response in cells pre-exposed with various background concentrations of EGF (x axis). (c) The maximal pAkt response in MCF10A cells exposed to different background doses of HGF (x axis) for 3 hr, followed by 2-, 4-, and 8-fold increase (different colors) of HGF. Inset shows experimental pAkt response over a wider range of background HGF levels. (d) The maximal pAkt responses (y axis) to HGF fold changes depended approximately logarithmically on the fold change (x axis). Maximal pAkt responses induced by stimulation with various HGF background levels (data points with the same shape and color) were combined and plotted as a function of the HGF fold change (x axis). Dashed line represents log-linear fit to data (Pearson's $r^2 = 0.88$, regression $p$ value $< 10^{-6}$). In all subpanels, error bars represent the standard deviation of n = 3 technical replicates. Source data: expt_data.mat (available in *Source code 1*).

The online version of this article includes the following figure supplement(s) for figure 6:

**Figure supplement 1.** Stimulation with different growth factors leads to specific removal of cognate receptors from the cell surface.

**Figure supplement 2.** Biological replicate of the experiment demonstrating relative sensing of HGF by pAkt (main text *Figure 6c,d*).

decreased HGF-induced responses (*Figure 6a*). Similarly, we observed a relatively small desensitization of HGF-induced responses due to pre-exposure with EGF, while there was a significant desensitization of EGF-induced pAkt responses (*Figure 6b*). We further confirmed that exposure of MCF10A cells to various concentrations of HGF leads to pronounced HGF-dependent removal of cMet from the cell surface, without significant removal of sEGFR (*Figure 6—figure supplement 1*). Similarly, the pre-exposure of cells to EGF leads to EGF-dependent removal of sEGFR without a significant change in surface cMet abundance (*Figure 6—figure supplement 1*). These observations support the mechanism in which the relative sensing of extracellular ligands relies on the memory of their past exposures effectively encoded in the abundances of their cognate cell-surface receptors.

Given the observed HGF-dependent removal of cell surface cMet receptors and the resulting pAkt desensitization, we investigated next whether the maximal pAkt response depends, similarly to EGF, on the relative fold changes in the level of extracellular HGF. To that end, we exposed cells to a range of different background levels of HGF, and then stimulated cells with different fold changes in HGF concentrations (*Figure 6c,d* and *Figure 6—figure supplement 2*). These experiments demonstrated that HGF-induced phosphorylation of Akt also depends primarily on the fold change in extracellular HGF concentration across almost an order of magnitude of background HGF exposures (between 0.1 and 1 ng/ml HGF) (*Figure 6c*). Moreover, like EGF, the maximum pAkt levels depended approximately log-linearly on the HGF fold change (*Figure 6d*).

Relative sensing of extracellular ligands should affect important downstream biological targets of the PI3K-Akt pathway. The FoxO3 transcription factor is a key effector of the pathway, and it is involved in diverse cellular processes including apoptosis, proliferation, and metabolism (*Webb and Brunet, 2014*). Akt phosphorylation of FoxO3 leads to its translocation from the nucleus to cytoplasm and subsequent transcriptional deactivation (*Webb and Brunet, 2014*). Notably, following Akt activation, the typical nuclear translocation timescale for FoxO family proteins is short (less than 5 min) (*Gross and Rotwein, 2017*). To investigate FoxO3 activation induced by EGF stimulation, we used quantitative immunofluorescence to measure its nuclear-to-cytoplasm ratio (*Worster et al., 2012*). We exposed cells to two different background EGF levels for three hours, and then treated them with two different abrupt fold changes in EGF concentrations. Consistent with relative sensing by pAkt, the nuclear-to-cytoplasmic ratio of FoxO3 also reflected the relative, rather than the absolute changes in EGF stimulation (*Figure 7* and *Figure 7—figure supplement 1*). Thus, relative sensing of the growth factor signal is faithfully transmitted in MCF10A cells to at least some of the physiologically important effectors of the PI3K-Akt pathway.

## Discussion

Receptor endocytosis and down-regulation, following ligand stimulation, has been canonically associated with signal and circuit desensitization (*Friedlander and Brenner, 2009*; *Sorkin and von Zastrow, 2009*; *Ferrell, 2016*). Our study suggests an additional and more quantitative role of receptors endocytosis in mammalian cells. Specifically, receptor endocytosis may allow cells to continuously monitor signals in their environment (*Becker et al., 2010*; *Brennan et al., 2012*; *Mitchell et al., 2015*) by dynamically adjusting the number of ligand-cognate receptors on the cell surface. Our analysis also demonstrates that the memory of past stimuli, effectively encoded in the number of surface receptors, may be signal-specific, at least for some ligands, due to the selective removal of ligand-cognate receptors. The combination of logarithmic pAkt response, and the logarithmic dependence of the ligand-specific memory on the background signal, allows cells to respond to relative changes in environmental stimuli. We note that the described relative sensing mechanism is not a direct consequence of either simple adaptation to various levels of background signals or logarithmic activation response (*Shoval et al., 2010*; *Adler et al., 2017*; *Adler and Alon, 2018*).

Previous studies (*Cohen-Saidon et al., 2009*; *Goentoro and Kirschner, 2009*; *Lee et al., 2014*) have demonstrated that transcriptional motifs may efficiently buffer cell-to-cell variability in signaling components when responding to a constant extracellular stimulation. In contrast, our study describes a non-transcriptional mechanism of sensing extracellular signal changes relative to past extracellular stimulation. Although the pAkt response to an abrupt stimulation is relatively fast (~5–15 min, *Figure 1a*), and therefore non-transcriptional in nature, the sustained production and delivery of cell surface receptors is essential to establishing the signal-dependent and receptor-mediated memory.

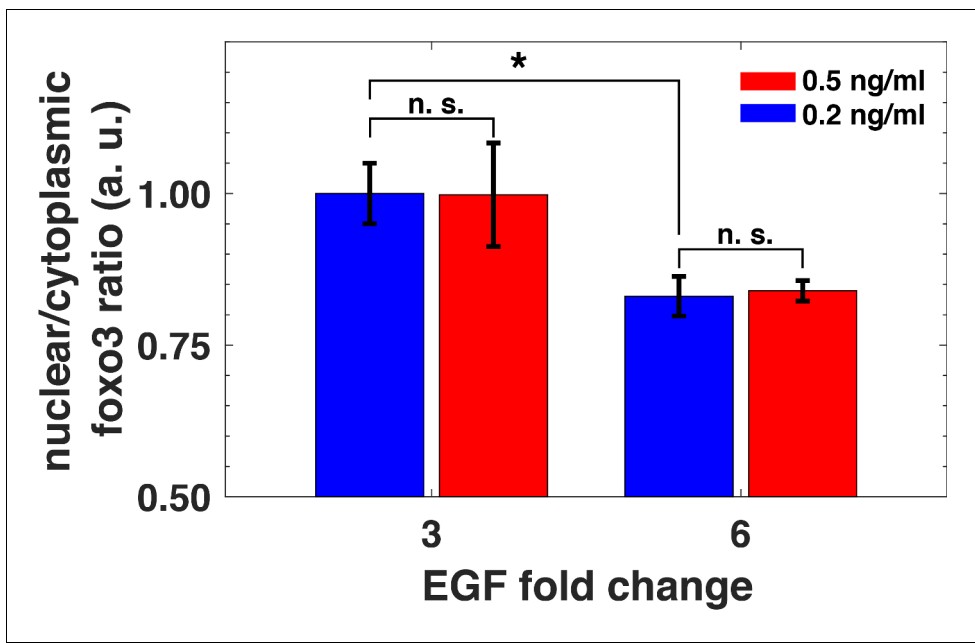

**Figure 7.** Relative sensing of EGF concentrations by pAkt is propagated to FoxO3. MCF10A cells were first exposed to two background concentrations of EGF for three hours, and then were stimulated with 3- and 6- fold increase in EGF concentrations. The ratio of nuclear-to-cytoplasmic FoxO3 levels (y-axis) was measured using quantitative immunofluorescence (Materials and methods) after 15 min of the EGF fold changes. Statistical significance was calculated using the Wilcoxon rank sum test (n = 5); * corresponds to p<0.01, and n. s. corresponds to p>0.1. Error bars represent the standard deviation of n = 5 technical replicates. Source data: expt_data.mat (available in *Source code 1*).

The online version of this article includes the following figure supplement(s) for figure 7:

**Figure supplement 1.** Biological replicate of the experiment showing relative sensing of EGF by FoxO3.

---

Therefore, sustained transcription and translation of network comonents are necessary for proper functioning of the described sensing mechanism.

Although there are usually $\sim 10^5$–$10^6$ EGFR receptors on mammalian cell surface (*Shi et al., 2016*), the downstream network response, for example Akt phosphorylation, often saturates when only a relatively small fraction (5–10%) of the receptors are bound to their cognate ligands (*Chen et al., 2009*; *Shi et al., 2016*). Our study suggests that one potential advantage of such a system architecture is that, beyond simple signal activation, it may endow cells with a large dynamic range of receptor abundances to memorize stimulation levels of multiple extracellular ligands (*Hart et al., 2013*; *Nandagopal et al., 2018*). Notably, signal-mediated removal has been reported for many other receptors and signaling systems, such as the G protein-coupled receptors (GPCRs) (*Ferguson, 2001*), involved in various sensory systems, and AMPA-type glutamate receptors (*Guskjolen, 2016*), implicated in synaptic plasticity. Therefore, similar relative sensing mechanisms may be important in multiple other receptor-based signaling cascades and across different biological contexts.

## Materials and methods

### Experimental methods

#### Measurement of EGF signaling responses

MCF10A cells were obtained from the ATCC and grown according to ATCC recommendations. Cell identity was confirmed by short tandem repeat (STR) profiling at the Dana-Farber Cancer Institute and cells were tested with the MycoAlert PLUS mycoplasma detection kit (Lonza) and found to be free of Mycoplasma prior to analyses. For experiments, 96 well plates (Thermo Scientific) were coated with type I collagen from rat tail (Sigma-Aldrich) by incubating plates with 65 μl of 4 mg/ml collagen I solution in PBS for 2 hr at room temperature, washed twice with PBS using a EL406

Microplate Washer Dispenser (BioTek), and then sterilized under UV light for 20 min prior to use. Cells were harvested during logarithmic growth and plated into collagen-coated 96 well plates using a EL406 Microplate Washer Dispenser. Cells were grown in 200 µl of complete medium for 24 hr, serum starved twice in starvation media (DMEM/F12 supplemented with 1% penicillin-streptomycin and 0.1% bovine serum albumin), incubated in 200 µl of starvation media for 19 hr, washed twice more, and incubated in 200 µl of starvation media for another hour. This time point constituted t = 0 for all experiments.

Treatment solutions were created by manual pipetting or by dispensing the appropriate amounts of epidermal growth factor (EGF, Peprotech), hepatocyte growth factor (HGF, Peprotech), or SC-79 (Sigma) into starvation media using a D300 Digital Dispenser (Hewlett-Packard). At t = 0 cells were stimulated with 100 µl of treatment solution and then incubated for the indicated times. For experiments requiring a second stimulus, cells were treated with an additional 100 µl of treatment solution at 3 hr and incubated for the indicated times. For fixation, 100 µl of supernatant were removed from the wells, replaced by 100 µl of 12% formaldehyde solution (Sigma) in phosphate buffered saline (PBS), and incubated for 30 min at room temperature.

All subsequent washes and treatments were performed with the EL406 Microplate Washer Dispenser. Cells were washed twice in PBS and permeabilized with 0.3% Triton X-100 (Sigma-Aldrich) in PBS for 30 min at room temperature (this step was omitted for measuring the surface expression of cMET and EGFR), washed once again in PBS, and blocked in 40 µl of Odyssey blocking buffer (LI-COR Biotechnology) for 60 min at room temperature. Cells were incubated with 30 µl of anti-phospho-Akt (Cell Signaling Technologies, 4060, 1:400), FoxO3 (Cell Signaling Technologies, 2497, 1:200), anti-Met (R and D Systems, AF276, 1:150), or anti-EGFR (Thermo Fisher Scientific, MA5-13319, 1:100) over night at 4℃. Cells were washed once in PBS and three times in PBS with 0.1% Tween 20 (Sigma-Aldrich; PBS-T for 5 min each and incubated with 30 µl of a 1:1000 dilution of secondary antibodies conjugated with Alexa Fluor 647 in Odyssey blocking buffer for 60 min at room temperature. Cells were washed two times in PBS-T, once with PBS, and stained for 30 min at room temperature with whole cell stain green (Thermo Fisher Scientific) and Hoechst (Thermo Fisher Scientific). Cells were washed three times in PBS, covered in 200 µl of PBS, and sealed for microscopy. Cells were imaged using an Operetta microscope (Perkin Elmer).

For the quantitative Western blots, about 70% confluent MCF10A cells were serum starved and treated with different concentrations of EGF (1, 0.56, 0.31, 0.18, 0.1 ng/mL). Cell lysate was prepared in Laemmli Sample Buffer (Bio-Rad) and subjected to SDS-PAGE in the 4–20% gradient gel (Bio-Rad). Western blots were performed using standard conditions with primary antibodies anti-phospho-EGFR (Cell Signaling Technologies, 3777, 1:1000) and anti-phospho-Akt (Cell Signaling Technologies, 4060, 1:1000) and anti-Actin (Santa Cruz Biotechnology, sc-47778 HRP, 1:5000). Secondary HRP-conjugated antibodies were acquired from Cell Signaling (7074, 1:10,000). Signals were detected with SuperSignal West Dura Extended Duration Substrate (Thermo Fisher Scientific) on a myECL Imager (Thermo Fisher Scientific) and analyzed by Image Studio Lite software (LI-COR Biosciences) by normalizing the signal from each antibody by the corresponding signal from Actin.

## Image Processing

Images were analyzed using the Columbus image data storage and analysis system (Perkin Elmer) to quantify single cell fluorescence measurements from each imaged well. The reported intensity values were obtained by first subtracting the background fluorescence of the well and subsequently the levels of pAKT at no stimulation at the same time. From each well we thus obtained a distribution of single cell measurements of a given target (pAkt, FoxO3, scMET or sEGFR). In each distribution we discarded the top and bottom 5% of points to remove outliers due to imaging and detection errors. The nuclear FoxO3 to cytoplasmic FoxO3 compartmentalization ratio was determined by the mean intensity in each area after image segmentation based on Hoechst and whole cell stain green at the single cell level. After that, we calculated the average of the resulting single cell distributions. For each condition, we performed multiple technical repeats (multiple wells), and as a final result reported the average of the corresponding single cell distribution averages and the associated standard deviations.

## Computational methods

In this section, we describe in detail (1) the model of the EGF/EGFR/Akt signaling pathway, (2) model assumptions, (3) model parameters and their bounds, (4) various relevant biological constraints that were imposed while fitting the model to the data, (5) the error function that was minimized in the parameter search, (6) the numerical procedure used to minimize the error function between model predictions and experimental data, and (7) in silico predictions.

### General structure of the computational model

The dynamic ODE model describing the EGF/EGFR signaling cascade leading to Akt phosphorylation (*Supplementary file 1a, b*, and *Equations A2–A20*) was based on the previous work by *Chen et al. (2009)*. We retained the components of the model relevant to EGF-dependent phosphorylation of EGFR and the subsequent cascade responsible for Akt phosphorylation. The resulting model consisted of 20 chemical species (see *Supplementary file 1a*) and was described by 24 parameters (20 reaction rate constants and four total species concentrations, *Supplementary files 1–3*, and *Equations A2– A20*).

The model included processes across three cellular compartments: cell surface (plasma membrane), cytoplasm, and endosomes. The model included interactions of the ligand with the receptors (ligand-binding and unbinding to receptor monomers and dimers) and subsequent receptor dimerization and undimerization. The model also included internalization of phosphorylated and unphosphorylated receptors, their recycling and degradation, phosphorylation and dephosphorylation by phosphatases.

### Main assumptions of the model

In agreement with available literature (*Wiley and Cunningham, 1982*; *Herbst et al., 1994*), we assumed that the rates of internalization, recycling, and degradation are different for inactive (unphosphorylated) and active (phosphorylated) receptors (*Supplementary file 1a*). We assumed that EGFR phosphatases in MCF10A cells are present at exceedingly high concentrations (*Kleiman et al., 2011*), and therefore we implemented the corresponding reaction of dephosphorylation of phosphorylated EGFRs (pEGFRs) as a first order reaction. We further assumed that activated receptors on plasma membrane and in endosomes are dephosphorylated by the phosphatase with the same rate (*Kleiman et al., 2011*).

We implemented PIP2 phosphorylation by pEGFR on the plasma membrane as a simplified effective first order process. Following the receptor-driven phosphorylation of PIP2 we retained the canonical signaling cascade of the PI3K/Akt activation (*Figure 2a* in the main text). We also implemented a first order reaction for action of the phosphatase on pAkt.

We assumed that cells are at steady state in terms of the abundances of ligand-free cell surface and endosomal receptors prior to ligand exposure. Specifically, prior to ligand exposure, the number of ligand-free EGFR monomers on cell surface and in endosomes, were derived based on the steady state condition of the corresponding equations.

In agreement with the literature (*Haugh and Meyer, 2002*; *Park et al., 2003*), we assumed that Akt can be phosphorylated only by cell-surface pEGFR, and not by endosomal pEGFR. Finally, we assumed that over the course of simulation extracellular ligand concentration remained constant, unless a step increase in EGF was applied. Here we refer to the background ligand stimulation as stimulation applied at time t = 0 to the cells that were previously not exposed to the ligand.

### Model parameters

The model parameters consisted of 4 total species abundances (PIP2, Akt, PDK1, and EGF receptors) and 20 rate constants. We collected multiple values of these parameters from literature (*Supplementary file 1a*). In our search for optimal rate parameters fitted to data, we allowed rate parameters to vary within half an order of magnitude from the lowest and the highest literature derived estimate (*Supplementary file 1a*). For parameters, for which experimental estimates were not available, we allowed up to four orders of magnitude in variation.

In addition, we allowed one and a half orders of variation in first order rate of EGF unbinding from receptors and the rate of EGFR phosphorylation in order to account for spatial organization of the receptors on the cell surface (*Mayawala et al., 2006*). We fixed the rate of pEGFR phosphatase

according to the measurement of this constant in MCF10A cells (*Kleiman et al., 2011*). In accordance with literature parameter estimates (*Supplementary file 1a*), we constrained the rate of ligand unbinding, receptor undimerization, receptor phosphorylation, and receptor dephosphorylation to be at least 10 times faster than receptor internalization (*Wiley et al., 1991*; *Herbst et al., 1994*; *Chen et al., 2009*; *Kleiman et al., 2011*).

Total number of EGFR receptors was limited to be between $10^5$–$10^6$ molecules per cell (*Niepel et al., 2013*). Total protein abundance of Akt was limited to be between $10^5$–$10^6$ molecules per cell (*Chen et al., 2009*). The abundance of PDK1 was limited between $10^3$–$10^6$ molecules per cell (*Chen et al., 2009*; *Wang et al., 2012*). Total abundance of lipid molecule PIP2 was limited between $10^{8.2}$–$10^{9.2}$. The abundance of PIP2 was calculated based on (1) surface area of MCF10A cells (calculated using a diameter of ~66 μm [*Imbalzano et al., 2009*] and assuming spherical cell shape), (2) total number of lipid molecules per 1 μm² of membrane (*Alberts et al., 1994*) (~$5 \times 10^6$), and (3) the fraction of PIP2 among all plasma membrane lipids (*Czech, 2000*) (0.75%); this corresponded to ~$5 \times 10^8$ molecules of PIP2 per cell.

## Additional constraints

In addition to the constraints imposed on network parameters directly through the experimentally measured data at EGF stimulations, we also required several additional constraints to better capture biology of EGFR signaling based on known literature. These constraints were either added as 'hard' constraints: parameter sets that did not agree with hard constraints were rejected, or as 'soft' constraints: parameter sets that did not agree with soft constraints were penalized using additional terms in the error function.

## Hard constraints

We constrained the total number of EGF receptors prior to EGF exposure to be between $10^5$–$10^6$ per cell, and surface EGFR to be within $10^5$–$10^6$ receptors per cell, in agreement with EGFR abundances reported for MCF10A cell lines (*Niepel et al., 2013*). In the model, the number of cell surface receptors was not a free parameter, but was calculated based on the steady state condition of the differential equations that describe the system.

## Soft constraint

We also implemented a 'soft' constraint that ensured realistic levels of phosphorylated Akt molecules. We required that at least 10% of total Akt gets phosphorylated at EGF doses close to Akt saturation, that is in our case 3.16 ng/ml EGF (*Chen et al., 2009*).

## Error function

The error function quantified the disagreement between model predictions and data and the soft constraints. At a given point $\theta$ in parameter space, we solved the system of ODEs describing EGF-induced Akt phosphorylation (see below) using the MATLAB and obtained model solutions $\{S_i\}$ for all experimentally measured conditions. Importantly, while the model predicts protein concentrations in units of number of molecules per cell, our experiments measured protein concentration up to a scaling factor. Therefore, we rescaled the model prediction using maximum likelihood linear-regression estimate (MLE) for both pAkt and sEGFR data between the model and the data respectively. Specifically, separately for pAkt and sEGFR, we fitted a linear model between the predictions from the ordinary differential equation model and the corresponding experimental measurements across multiple EGF doses and time points. We rescaled the predictions based on the slope and the intercept of the linear model fit. The scaled predictions were used in the evaluation of the error.

The error function comprised of two different contributions. The first term was defined as the sum of the squared differences between the model predictions $\{S_i\}$ at parameter value $\theta$ and the corresponding experimental data D, taking into account corresponding experimental errors σ s (*Equation S1a*). Next, we imposed the soft constraints described above as squared error terms (*Equation S1b*, and *Supplementary file 1b* for species abbreviation). The total error function was the sum of these two contributions (*Equation S1c*).

$$\hat{E}_1(\theta) = \sum_{k=1}^{n} \frac{(S_k(\theta) - D_k)^2}{2\sigma_k^2} \tag{S1a}$$

$$\hat{E}_2(\theta) = \frac{(f_{akt,max} - 0.1)^2}{2 \times (0.004)^2} \; if \; f_{akt,max} < 0.1 \tag{S1b}$$

$$\hat{E}(\theta) = \hat{E}_1(\theta) + \hat{E}_2(\theta) \tag{S1c}$$

where

$$f_{akt,max} = \frac{C19_{max}}{C17} \tag{S1d}$$

The standard deviation 0.004 in *Equation S1b* was chosen to ensure that the maximum pAkt levels were guaranteed to be above 10% of total Akt levels. Lower values lead to a very high rejection rate in the simulated annealing procedure and higher values were likely to return parameter points that did not satisfy the constraint that maximum pAkt levels were at least 10% of total Akt levels. The active endocytosis and degradation of cell surface receptors in our system occurred mostly between EGF doses of 0 ng/ml and 3.16 ng/ml. Accordingly, we fit the model using experimental data collected in the same range of EGF stimulations.

The error function in *Equation 1 a* contained the following experimentally measured data points: pAkt time courses measured up to 180 min (5, 10, 15, 30, 45, 90, and 180 min) across range of EGF doses between 0.03 and 3.16 ng/ml (0.03,0.1,0.3,1,3.16 ng/ml EGF) and sEGFR levels at 2.5 and 3 hr across a range of EGF stimulation doses (0 ng/ml and 0.03–3.16 ng/ml).

In *Equation 1 a*, index $k$ runs through all $n$ experimentally measured data points (5 doses x 7 time points = 35 total points) and sEGFR measurements (6 doses x 2 time points = 12 total points).

Overall, the error function had a total of 50 terms (35 pAkt measurements, 12 sEGFR measurements, and one soft constraint). We minimized this error by searching through the parameter space using simulated annealing (SA) described in the next section.

## Simulated Annealing optimization

Given that the mechanistic ODE models constrained by experimental measurements of several dynamical quantities are usually underdetermined (*Chen et al., 2009*), we used simulated annealing (SA) (*Kirkpatrick et al., 1983*), to numerically search the model's parameter space.

The overall error (*Equation S1c*) was minimized with SA in order to determine parameter sets that are most consistent with the experimental measurements. Following standard SA optimization scheme, we ran a random walk in the model's parameter space. At each point in the parameter space we accept or reject a next proposed parameter set according to the Metropolis criterion and a likelihood that is the negative exponential of the error function in *Equations S1a*. Following a conventional SA protocol, we used an additional parameter, temperature, which allowed steps with relatively large change in the likelihood score to explore large parameter space. The temperature was decreased gradually to find a local minimum of the likelihood function.

To find multiple parameter sets that fit the experimental data, we ran 100 independent SA chains with randomly selected starting points spread out across allowed parameter ranges. Each chain was started at high temperature and was cooled down in 12 stages to the lowest temperature (using the sequence of temperatures: 400, 200, 100, 50, 20, 10, 5, 2, 1, 0.5, 0.25, 0.1). At each temperature, 1500 steps in parameter space were performed. In each step, on an average four randomly chosen parameters (out of the 24) were changed in order to speed up the search in the parameter space.

## Predictions from SA

For individual chains, the parameter set with the lowest error was recorded. The averages of parameter values from the 10 best-fit chains are shown in *Supplementary file 1a*. We used these top 10 optimized parameter sets (*Supplementary file 1a*, *Figure 2—figure supplement 1*) to explore phenomenon of relative sensing in silico. For each parameter set, we simulated the following. The model

was first exposed to the background EGF concentration for 50 hr to ensure that all species reached a steady state. The model was subsequently exposed to a step increase in EGF concentration (2-, 3-, 4-, or 6- fold). After the step increase, EGF was kept constant as well. We noted the maximum Akt phosphorylation level at each background concentration and EGF fold-change. For each fit, we obtained a series of maximum pAkt responses across different initial EGF concentrations and for multiple fold changes as well as time integrals of pAkt responses between 0 and 30 min. We combined the predicted relative sensing dose responses at every background EGF level and at every fold by taking the average (and the corresponding standard deviation) across predictions from all 10 best parameter sets. We then plot the resulting dose response as seen in *Figure 2c,d* of the main text.

## Statement of source code availability

All data and source code are available at: https://github.com/dixitpd/FoldChange (*Dixit, 2020*; copy archived at https://github.com/elifesciences-publications/FoldChange).

---

# Additional information

### Funding

| Funder | Grant reference number | Author |
| --- | --- | --- |
| National Institutes of Health | R01CA201276 | Eugenia Lyashenko<br>Purushottam D Dixit<br>Dennis Vitkup |
| National Institutes of Health | U54CA209997 | Eugenia Lyashenko<br>Purushottam D Dixit<br>Dennis Vitkup |
| National Institutes of Health | U54HL127365 | Mario Niepel<br>Sang Kyun Lim<br>Peter K Sorger |
| National Institutes of Health | U54CA225088 | Peter K Sorger |

The funders had no role in study design, data collection and interpretation, or the decision to submit the work for publication.

### Author contributions

Eugenia Lyashenko, Conceptualization, Data curation, Software, Formal analysis, Investigation, Visualization, Methodology; Mario Niepel, Conceptualization, Resources, Data curation, Formal analysis, Validation, Investigation, Visualization, Methodology, Experiments; Purushottam D Dixit, Conceptualization, Data curation, Software, Formal analysis, Validation, Investigation, Visualization, Methodology; Sang Kyun Lim, Resources, Data curation, Formal analysis, Methodology, Experiments; Peter K Sorger, Resources, Supervision, Funding acquisition, Methodology; Dennis Vitkup, Conceptualization, Supervision, Funding acquisition, Investigation, Methodology, Project administration

### Author ORCIDs

Eugenia Lyashenko (iD) https://orcid.org/0000-0002-6538-9462
Mario Niepel (iD) https://orcid.org/0000-0003-1415-6295
Purushottam D Dixit (iD) https://orcid.org/0000-0003-3282-0866
Peter K Sorger (iD) http://orcid.org/0000-0002-3364-1838
Dennis Vitkup (iD) https://orcid.org/0000-0003-4259-8162

### Decision letter and Author response

Decision letter https://doi.org/10.7554/eLife.50342.sa1
Author response https://doi.org/10.7554/eLife.50342.sa2

## Additional files

### Supplementary files

• Source code 1. Immunofluorescence data and source code for fitting the ODE model to the data and further simulations.

• Supplementary file 1. Table of ODE model parameter ranges. Descriptions and literature-based estimates of rate parameters and species abundances.

• Supplementary file 2. Description of chemical species in the ODE model.

• Supplementary file 3. Description of chemical reactions in the ODE model.

• Transparent reporting form

### Data availability

All data used in this study and the code used for simulations is available at https://github.com/dix-itpd/FoldChange (copy archived at https://github.com/elifesciences-publications/FoldChange).

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

**Appendix 1**

## Analytical model of relative sensing

Below we present the analytical model to understand the biochemical mechanisms underlying the observed relative sensing in the EGFR/Akt pathway. The model focuses on the receptors level signaling. Since Akt phosphorylation is directly downstream of receptor phosphorylation, the conclusions of the model hold for Akt phosphorylation as well.

In section 1 we show that after continuous exposure to an activiating ligand with concentration $[L]_0$, the steady-state level of cell-surface receptors $[R]_T$ decreases approximately as the negative logarithm of the ligand concentration, that is $[R]_T \sim (constant - \log [L]_0)$. In this way, the steady state abundance of cell surface receptors effectively encodes the memory of the background ligand concentration. In section 2, we show that the phosphorylation response induced by an increase in the extracellular ligand concentration from $[L]_0$ to $[L]_1$ depends approximately logarithmically on the ligand concentration and linearly on the current abundance of cell-surface receptors $[R]_T$. Based on results derived in the secton 1 and section 2, we then demonstrate that the phosphorylation response of the signaling network to an increase in ligand concentration primarily depends on the relative fold increase in ligand concentration ($[L]_1/[L]_0$). In section 3, we use the analytical model to identify two dimensionless aggregate parameters ($\alpha$ and $\beta$) that determine the ranges of background ligand concetation where relative sensing can be observed. In section 4 we derive and analyze a similar analytical model for a signal transduction network in which the phopshorylation signal is initiated by monomeric receptors.

## Main assumptions of the model

In the simplified analytical model, following experimental evidence (*Huang et al., 2016*; *Macdonald and Pike, 2008*) the prevailing mode of EGFR dimerization in the range of extracellular EGF concentrations that we consider is assumed to be one EGF-bound EGFR monomer bound to one ligand-free EGF receptor. While phosphorylated EGFRs can signal from both the plasma membrane and endosomes, some components of the PI3K/Akt signaling pathway are largely restricted to the plasma membrane. Consequently, Akt phosphorylation occurs primarily from the phosphorylated EGFRs localized at the plasma membrane (*Haugh and Meyer, 2002*; *Park et al., 2003*). Thus, we assume that Akt molecules cannot get phosphorylated from phosphorylated EGFRs localized in the endosomes. We also assume that endosomal EGFR-bound EGF ligands do not dissociate from the receptors (*Reddy et al., 1998*). In the endosomes the model only allows receptor phosphorylation/dephosphorylation, recycling to plasma membrane, and degradation. We assume that the rates of phosphorylation and dephosphorylation are the same for cell surface receptors as well as internalized receptors. We also assume that rates of internalization, recycling, and degradation of phosphorylated (activated) receptors are different from those of the non-phosphorylated (non-activated) receptors. Finally, we assume that the extracellular ligand concentration remains constant at background stimulation and after the step increase in ligand concentration.

## Section 1. steady state level of surface receptors

First, we calculate the level of cell surface receptors at steady state when cells are exposed to a constant stimulus with ligand concentration $[L]_0$. Given the aforementioned assumptions, *Equations A1-A8* describe the dynamics of receptor signaling cascade in presence of ligand at concentration $[L]_0$ (*Appendix 1—figure 1*). In *Equations A1-A8* below, we use * to denote phosphorylated receptors and the subscript $i$ to denote endosomal receptors.

$$\frac{d[R]}{dt} = k_{prod} - k_1 [L]_0 [R] + k_{-1}[LR] - k_2[R][LR] + k_{-2}[LR_2] - k_i[R] + k_{rec}[R_i] \tag{A1}$$

$$\frac{d[LR]}{dt} = k_1 [L]_0 [R] - k_{-1}[LR] - k_2[R][LR] + k_{-2}[LR_2] - k_i[LR] + k_{rec}[LR_i] \tag{A2}$$

$$\frac{d[LR_2]}{dt} = k_2[R][LR] - k_{-2}[LR_2] - k_p[LR_2] + k_{dp}[LR_2^*] - k_i[LR_2] + k_{rec}[LR_{2i}] \tag{A3}$$

$$\frac{d[LR_2^*]}{dt} = k_p[LR_2] - k_{dp}[LR_2^*] - k_i^*[LR_2^*] + k_{rec}^*[LR_{2i}^*] \tag{A4}$$

$$\frac{d[R_i]}{dt} = k_i[R] - k_{rec}[R_i] - k_{deg}[R_i] \tag{A5}$$

$$\frac{d[LR_i]}{dt} = k_i[LR] - k_{rec}[LR_i] - k_{deg}[LR_i] \tag{A6}$$

$$\frac{d[LR_{2i}]}{dt} = k_i[LR_2] + k_{dp}[LR_{2i}^*] - k_p[LR_{2i}] - k_{rec}[LR_{2i}] - k_{deg}[LR_{2i}] \tag{A7}$$

$$\frac{d[LR_{2i}^*]}{dt} = k_i^*[LR_2^*] + k_p[LR_{2i}] - k_{dp}[LR_{2i}^*] - k_{rec}^*[LR_{2i}^*] - k_{deg}^*[LR_{2i}^*] \tag{A8}$$

- *Equation A1* describes the dynamics of the concentration $[R]$ of ligand-free receptors. $k_{prod}$ is the rate of delivery of free receptors to cell surface, $k_1$ is the rate constant of ligand binding to receptors, $k_{-1}$ is the rate constant of ligand unbinding from the ligand-receptor complex, $k_2$ is the rate of receptor dimerization, $k_{-2}$ is the rate of receptor undimerization, $k_i$ is the basal internalization rate of non-activated receptors, and finally, $k_{rec}$ is the rate of recycling of internalized unphosphorylated receptors to cell surface.
- *Equation A2* describes dynamics of the concentration $[LR]$ of the ligand bound non-activated monomeric receptors.
- *Equation A3* describes the dynamics of the concentration $[LR_2]$ of the unphosphorylated dimer species. $k_p$ is the rate of receptor phosphorylation and $k_{dp}$ is the rate of receptor dephosphorylation.
- *Equation A4* describes the dynamics of the concentration $[LR^*_2]$ of the phosphorylated dimer species. $k_i^*$ is the internalization rate of phosphorylated receptors and $k_{rec}^*$ is the rate of recycling of phosphorylated receptors.
- *Equation A5* describes the dynamics of the concentration $[R_i]$ of the internalized ligand-free receptor monomers. $k_{deg}$ is the rate of degradation of unphosphorylated receptors.
- *Equation A6* describes the dynamics of the concentration $[LR_i]$ of the internalized ligand-bound non-activated receptor monomers.
- *Equation A7* describes the dynamics of the concentration $[LR_{2i}]$ of the internalized unphosphorylated receptor dimers.
- *Equation A8* describes the dynamics of the concentration $[LR^*_{2i}]$ of the internalized phosphorylated receptors. $k^*_{deg}$ is the rate of degradation of phosphorylated receptors.

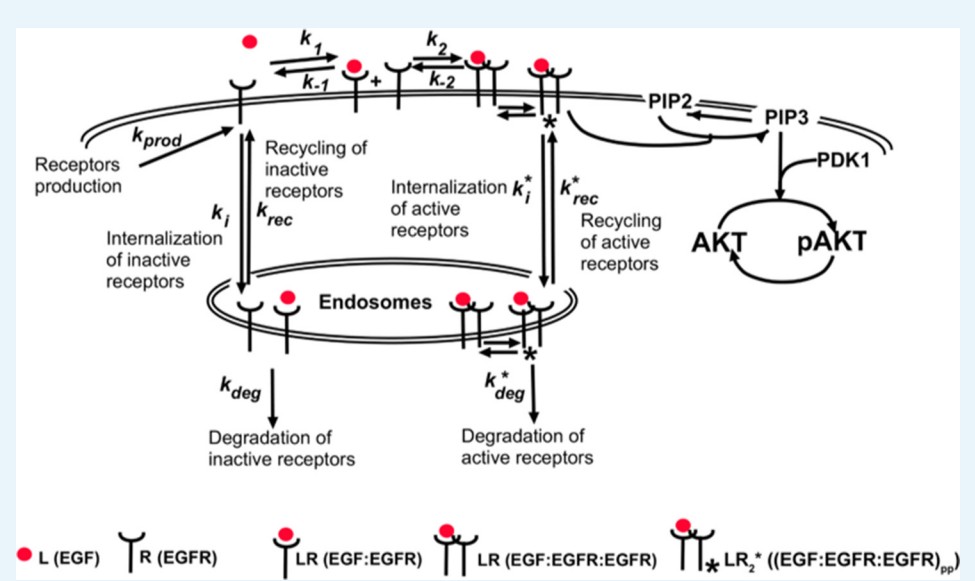

**Appendix 1—figure 1.** Model schematics of dimer-activated receptors signaling cascade.

In our calculations below we consider all concentrations (except for the ligand) to be measured on a per cell basis. For convenience we also introduce the following constants and equations:

$$R_0 = \frac{k_{prod}}{k_i} \times \frac{k_{deg} + k_{rec}}{k_{deg}} \tag{A9}$$

steady state abundance of surface receptors in the absence of ligand, $[L]=0$

$$[R]_T = [R] + [LR] + 2[LR_2] + 2\left[LR_2^*\right] \tag{A10}$$

the total cell-surface receptor level at steady state

$$K_{d1} = \frac{k_{-1}}{k_1} \tag{A11}$$

the equlibrium dissociation constant of ligand binding to receptors.

$$K_{d2} = \frac{k_2}{k_{-2}} \tag{A12}$$

the equilibrium constant of the undimerization reaction.

$$u_0 = \frac{[L]_0}{K_{d1}} \tag{A13}$$

the ligand concentration relative to the ligand-receptor equilibrium dissociation constant

We note that $K_{d1}$ has the units of moles and $K_{d2}$ has the units of molecules per cell (same as $R_0$), and $u_0$ is a dimensionless quantity.

As has been previously reported (**Wiley and Cunningham, 1982**) and as was also observed in our experimental system, surface EGFR levels reach quasi-steady state within hours of continuous stimulation with EGF (main text **Figure 1c**). Importantly, ligand unbinding, receptors undimerization, and receptor phosphporylation-dephosphprylation happen at a substantially faster rate than basal receptor internalization (**Chen et al., 2009**; **Kleiman et al., 2011**). Moreover, when we substitute the average parameter values from the fits to the computational model (main text and Methods), we have:

$$\frac{k_{-1}}{k_i} \sim \frac{0.3\ s^{-1}}{1.5 \times 10^{-4} s^{-1}} \sim 2000 \gg 1,$$

$$\frac{k_{-2}}{k_i} \sim \frac{0.05\ s^{-1}}{1.5 \times 10^{-4} s^{-1}} \sim 380 \gg 1,$$

$$\frac{k_p}{k_i} \sim \frac{1\ s^{-1}}{1.5 \times 10^{-4} s^{-1}} \sim 7000 \gg 1,$$

$$\frac{k_{dp}}{k_i} \sim \frac{0.07\ s^{-1}}{1.5 \times 10^{-4} s^{-1}} \sim 480 \gg 1$$

(A14)

From **Equations A1-A8** and the limits imposed by inequalities A14, we calculate the receptor level at steady state with continuous exposure to the ligand by setting the rates of change of individual concentrations to zero. Based on the available knowledge about the biology of EGF receptors and parameters of our model (**Equations A14**), we neglect the internalization and recycling of ligand-bound monomers and unphosphorylated dimers. Consequently, we neglect degradation of ligand-bound receptor monomers and unphosphorylated receptor dimers. As a result, we assume that at steady state, the fraction of monomeric ligand bound receptors in the endosomes is negligible. We have

$$0 = k_{prod} - k_1[L]_0[R] + k_{-1}[LR] - k_2[R][LR] + k_{-2}[LR_2] - k_i[R] + k_{rec}[R_i]$$

(A15)

$$0 = k_1[L]_0[R] - k_{-1}[LR] - k_2[R][LR] + k_{-2}[LR_2]$$

(A16)

$$0 = k_2[R][LR] - k_{-2}[LR_2] - k_p[LR_2] + k_{dp}[LR_2^*]$$

(A17)

$$0 = k_p[LR_2] - k_{dp}[LR_2^*] - k_i^*[LR_2^*] + k_{rec}^*[LR_2^*i]$$

(A18)

$$0 = k_i[R] - k_{rec}[R_i] - k_{deg}[R_i]$$

(A19)

$$0 = k_{dp}[LR_{2_i}^*] - k_{deg}[LR_{2_i}] - k_p[LR_2 i]$$

(A20)

$$0 = k_p[LR_2 i] - k_{dp}[LR_2 i^*] + k_i^*[LR_2^*] - k_{rec}^*[LR_2 i^*] - k_{deg}^*[LR_2 i^*]$$

(A21)

Solving for steady state **Equations A15-A21** when ligand concentration is $[L]_0 = u_0{}^*K_{d1}$, we obtain the formula for the total number of receptors on the cell surface,

$$[R]_T = 2R_0 \left[ \frac{1 + u_0}{1 + \sqrt{1 + 8\alpha\beta u_0}} + \frac{2\alpha u_0}{1 + 4\alpha\beta u_0 + \sqrt{1 + 8\alpha\beta u_0}} \right] \approx \frac{2R_0}{1 + \sqrt{1 + 8\alpha\beta u_0}}$$

(A22)

where we have introduced two aggregate parameters that we denote $\alpha$ and $\beta$ (see **Equations A23a and A23**b below).

The first term in **Equation A22** represents the concentration of monomeric surface receptors (ligand-free $[R]$ and ligand-bound monomeric receptors $[LR]$) at steady state, the second term represents the steady state concentration of dimeric receptors (unphosphorylated dimers $[LR_2]$ and phosphorylated receptors $[LR^*_2]$). When the ligand concentration is much lower than the equilibrium dissociation constant; $u_0 \ll 1$, we can assume that the majority of cell surface receptors are ligand free as indicated in the second approximation in **Equation A22**.

Remarkably, even though **Equations A1-A8** are governed by more than ten rate parameters, the total concentration of receptors on the surface $[R]_T$ at steady state depends only on two composite aggregate parameters α and β and the ligand concentration $u_0$. The two aggregate parameters are defined as,

$$\alpha = \frac{k_{dp} + k_p}{k_{dp}} \times \frac{R_0}{K_{d2}},$$

(A23a)

$$\beta = \frac{k_p}{k_{dp} + k_p} \times \frac{k_i^*}{k_i} \times \frac{\frac{k_{deg}^*}{k_{deg}^* + k_{rec}^*}}{\frac{k_{deg}}{k_{deg} + k_{rec}}}$$

(A23b)

The parameter $\alpha$ quantifies the ligand-sensitivity of the signaling system, an increase in the value of $\alpha$ leads to a higher signal sensitivity and an increase in receptor dimerization and

phosphorylation. The parameter $\beta$ quantifies the combination of two biases leading to preferential internalization and degradation of phosphorylated receptors. An increase in the value of $\beta$ leads to a larger fraction of active (phosphorylated) receptors being internalized and degraded. The first term in A23b is the fraction of dimeric receptors that are phosphorylated. The second term $k^*_i/k_i$ is the relative rate at which phosphorylated receptors are trafficked to the endosomes compared to unphosphorylated receptors. Finally, the third term quantifies the ratio of the two following fractions; $k^*_{deg}/(k^*_{rec} + k^*_{deg})$ is the fraction of phosphorylated receptors that are transported from endosomes to be degraded and $k_{deg}/(k_{rec} + k_{deg})$ is the fraction of unphosphorylated receptors that are transported from endosomes to be degraded.

Using average values of parameters from our dynamical ODE model fits (Materials and methods and **Supplementary files 1, 2, 3**), we estimate $\alpha \sim 16$ and $\beta \sim 40$. From here onwards, we use these two values of $\alpha$ and $\beta$ in our analysis. In **Appendix 1—figure 2** we plot the steady state cell surface EGFR levels predicted by **Equation A22** as a function of the continuous background ligand stimulation $u_0$ for these values of $\alpha$ and $\beta$. Notably, the receptor levels decrease approximately logarithmically as a function of $u_0$ over nearly two orders of magnitude in ligand concentration between $u_0 \sim 10^{-3.5} - 10^{-2}$.

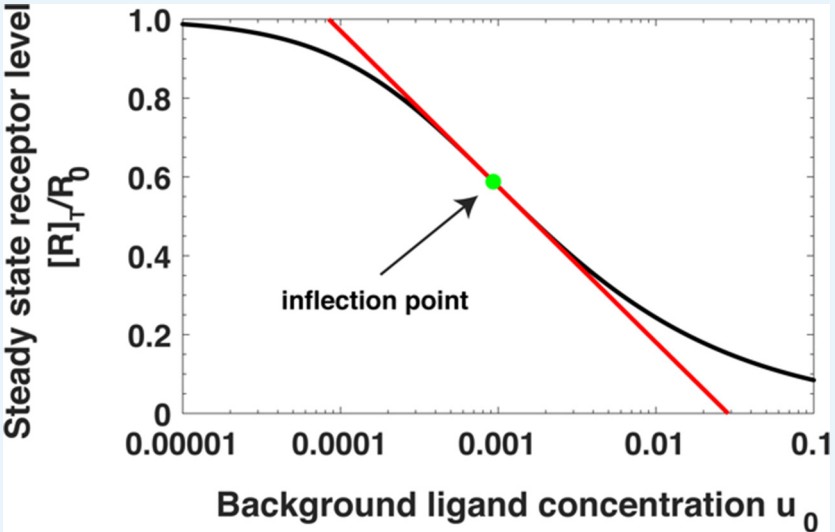

**Appendix 1—figure 2.** Steady state receptor levels decrease proportional to the logarithm of ligand concentration. The steady state cell surface receptor level $[R]_T$ for receptors (black line) given by **Equation (A22)** depends approximately on the logarithm of the ligand concentration $u_0 = [L]_0/K_{d1}$ (red line, **Equations A24**). The inflection point is indicated by a green dot.

The logarithmic dependence can be quantified as follows. $[R]_T$ in **Equation A22** has an inflection point with respect to the logarithm of the ligand concentration; $\frac{d^2[R]_T}{d \log u_0^2} = 0$ at $u_0 = u_{flex} = \frac{1+\sqrt{2}}{4\alpha\beta}$. Near the inflection point, $[R]_T$ can be approximated as a log-linear function of $u_0$. Expanding **Equation A22** near $u_0 = u_{flex}$, we have

$$r = \frac{[R]_T}{R_0} \approx constant - a \log u_0 \qquad (A24)$$

where $a$. Notably, $a = 3 - 2\sqrt{2}$ does not depend on $\alpha$ and $\beta$. Here $r$ is the fraction of receptors remaining on the cell surface.

## Section 2. Relative sensing of extracellular ligand by the signaling cascade

In this section, using a surface-receptor level analytical model, we show that receptor phosphorylation can sense relative changes in extracellular ligands. Based on the equation for the total number of receptors at the steady-state obtained in the previous section, we calculate here the maximum of the EGFR phosphorylation level in response to an instant change in EGF concentration from a background level $[L]_0$ to a new level $[L]_1$.

Main assumptions of the model. We assume that the dynamic maximum of the phosphorylation response is reached through the rapid equilibration between ligand binding and unbinding and receptor phosphorylation and dephoshorylation and assume no contribution from relatively slow receptor internalization (*Kleiman et al., 2011*). Consequently, at the time phosphorylated EGFRs reach their maximum level, the total number of receptors on the cell surface, $[R]_T$, is approximately similar to the steady state level determined by the pre-exposure to ligand concentration of $[L]_0$. With these approximations, we have at pseudo-steady state conditions at the maximum pEGFR activity:

$$0 = -k_1[L]_1[R] + k_{-1}[LR] - k_2[R][LR] + k_{-2}[LR_2] \tag{A25}$$

$$0 = k_1[L]_1[R] - k_{-1}[LR] - k_2[R][LR] + k_{-2}[LR_2] \tag{A26}$$

$$0 = k_2[R][LR] - k_{-2}[LR_2] - k_p[LR_2] + k_{dp}[LR_2^*] \tag{A27}$$

$$0 = k_p[LR_2] - k_{dp}[LR_2^*] \tag{A28}$$

$$[R]_T = [R] + [LR] + 2[LR_2] + 2[LR_2^*] \tag{A29}$$

Solving *Equations A25, A26, A27, A28, A29* for maximum pEGFR level of active receptors $[LR_2^*]$, we get

$$[LR_2^*] = \frac{k_p R_0}{k_{dp} + k_p} \times r \left[ \frac{1}{2} - \frac{1}{1 + \sqrt{1 + \frac{8\alpha u_1}{(1+u_1)^2} \times r}} \right] \approx \frac{k_p R_0}{k_{dp} + k_p} \times r \left[ \frac{1}{2} - \frac{1}{1 + \sqrt{1 + 8\alpha u_1 r}} \right] \tag{A30}$$

The leading term $\frac{k_p R_0}{k_{dp}+k_p}$ in *Equation A30* is a constant that determines only the overall strength of the response. From here onwards, we neglect it. Note that in *Equation A30*, the total receptor concentration $r$ depends on the background ligand concentration $[L]_0$ before a step increase in the ligand concentration (see *Equation A22*). The result of substituting the expression for the receptor level (*Equation A22*) in *Equation A30* cannot be written down as a simple analytical expression. In *Appendix 1—figure 3*, we numerically examine how $[LR_2^*]$, given by *Equation A30*, depends on the fold change in ligand concentration, that is $[L]_1/[L]_0$, at different background ligand concentrations $[L]_0$. Notably, $[LR_2^*]$ mostly depends on the ratio $[L]_1/[L]_0$ and only weakly on the background exposure levels $[L]_0$ over an order of magnitude in background ligand concentration.

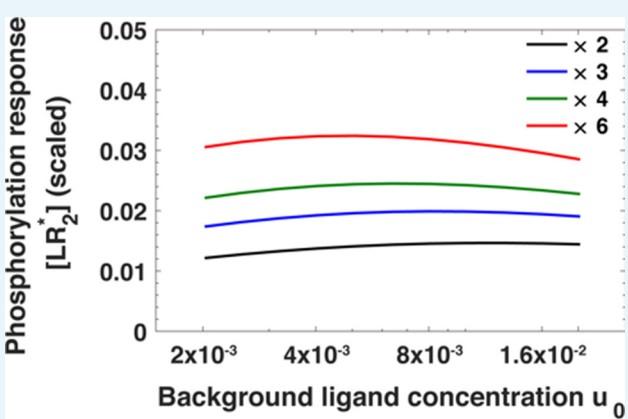

**Appendix 1—figure 3.** The maximum of the EGFR phosphorylation response $[LR^*_2]$ as a function of ligand concentration $u_0$ before step change in ligand concentration for fold changes $u_1/u_0$ in ligand concentration ranging from $u_1/u_0 = 2$ (black) to $u_1/u_0 = 6$ (red). The phosphorylation response is estimated using **Equation A30**.

The relative sensing phenomenon can be intuitively understood by considering an approximation to Equation A30 that allows us to quantify the contribution of ligand stimulation and steady state receptor levels in maximum phosphorylated EGFR response. As seen above, for a given background concentration $r$ the corresponding steady state receptor level $u_0 = u_{flex}$ decreases approximately as the logarithm of the background concentration near $u_0 = u_{flex}$ of the relationship between ligand concentration and steady state receptor level (**Equation A22**). When $u_0 = u_{flex}$, $r = r_{flex} = 2 - \sqrt{2}$. Similarly, the phosphorylation response $[LR^*_2]$ upon a step change in ligand concentration from $u_0$ to $u_1$ can also be approximated as a log-linear function of the ligand concentration $u_1$ near $u_1 = \frac{1+\sqrt{2}}{4r\alpha}$. Here $r$ is the steady state cell surface EGFR levels after exposing the cells to a background ligand concentration $u_0$. To understand how $[LR^*_2]$ depends separately on $r$ and $u_1$, we expand using, Taylor series, $[LR^*_2]$ given by **Equation A30** linearly in $r$ and log-linearly in $u_1$ near $r = r_{flex}$ and $u_1 = \frac{1+\sqrt{2}}{4r_{flex}\alpha}$. We have

$$[LR^*_2] \approx constant + b \log u_1 + c \times r \tag{A31}$$

where $b = \left(5 - \frac{7}{\sqrt{2}}\right)$ and $c = \left(1 - \frac{1}{\sqrt{2}}\right)$.

In **Appendix 1—figure 4**, we show a direct comparison between the phosphorylation response as predicted by **Equation A30** as well as its approximation given by **Equation A31**. We evaluate the phosphorylation response at $r = r_{flex}$ by varying the ligand concentration $u_1$ between $u_1 = 10^{-2}$-$10^{-0.75}$. The x-coordinate corresponds to the phosphorylation response predicted by **Equation A30** and the y-coordinate corresponds to the phosphorylation response predicted by **Equation A31**. Individual data points are represented as black circles. The dashed red line represents the line y = x. From **Appendix 1—figure 4**, it is clear that **Equation A31** agrees very well with **Equation A30** over a broad range of variation in ligand concentrations.

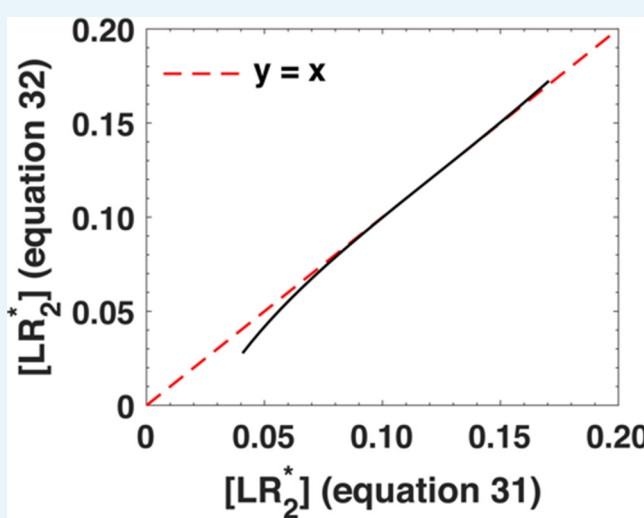

**Appendix 1—figure 4.** Comparison between the phosphorylation response as predicted by *Equation A30* (x-axis) and its approximation, *Equation A31* (y-axis). We varied the ligand concentration $u_1$ between $u_1 = 10^{-2}$-$10^{-0.75}$. The black dots represent individual data points and the dashed red line represents x = y line.

Substituting the log-linear relationship between $r$ and the background ligand concentration $u_0$ given by *Equation A24* we have

$$[LR_2^*] \approx constant + b\log u_1 + c \times r \approx constant + b\log u_1 - c \times a\log u_0 \tag{A32}$$

$$\Rightarrow [LR_2^*] \approx constant + b\log\frac{u_1}{u_0} \tag{A33}$$

*Equation A33* follows from *Equation A32* because

$$a \times c = \left(3 - 2\sqrt{2}\right) \times \left(1 - \frac{1}{\sqrt{2}}\right) = \left(5 - \frac{7}{\sqrt{2}}\right) = b \tag{A34}$$

Consequently, the phosphorylation response after a step change in ligand concentration mostly depends on the ratio of ligand concentrations rather than on the background ligand concentration.

## Section 3. Aggregate network parameters determine the range of relative sensing

In this section we explore whether the observed relative sensing is a result of *fine-tuning* of rate parameters in the EGF/EGFR pathway or if it is a robust phenomenon that is relatively insensitive to parameter variations.

The analysis of the model shows that the maximum phosphorylation response $[LR^*_2]$ (*Equation A30*) primarily depends on two aggregate parameters $\alpha$ and $\beta$ (*Equations A23a and A23b*). In order to compare phosphorylation response at different values of $\alpha$ and $\beta$, we define the scaled phosphorylation response $[LR^*_2]$ (scaled) by normalizing $[LR^*_2]$ by its maximum over the background ligand concentration $[L]_0$ at a fixed value of $\alpha$ and $\beta$. Such a scaled phosphorylation response varies between zero and one for all values of $[L]_0$, $[L]_1$, $\alpha$, and $\beta$.

Using as an example a fold change of 6, $[L]_1 = 6 \times [L]_0$, in *Appendix 1— figure 5* we show $[LR^*_2]$ (scaled) as a function of background ligand concentration at different values of $\alpha$ (panel a) and $\beta$ (panel b). The different colors represent different accuracy values of relative sensing. For example, the color red represents the range of background ligand concentrations over which the phosphorylation response is within 90% of the maximum (10% deviation). The color yellow represents the range of background ligand concentrations over which phosphorylation

response is within 80% of the maximum (20% deviation) and so on. Vertical green lines indicate the region of 90% accuracy; the phosphorylation response $[LR^*_2]$ (scaled) varies within 10% of the maximum. Notably, as $\alpha$ increases, corresponding to a larger fraction of receptors that get phosphorylated, the span of background concentrations over which signaling network exhibits relative sensing increases as well. In panel b), we show $[LR^*_2]$ (scaled) as a function of background ligand concentration at different values of $\beta$. As $\beta$ increases, the range of background concentrations over which relative sensing holds increases as well. The green arrows show the range of background ligand concentrations over which relative sensing holds for the values of $\alpha$ and $\beta$ evaluated from the model fits to the data.

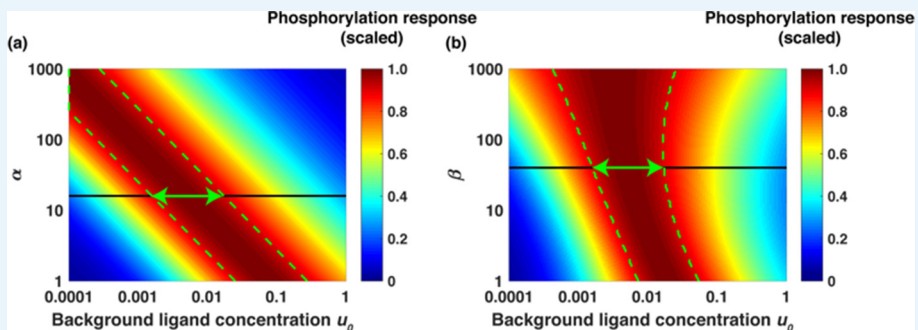

**Appendix 1—figure 5.** Dimensionless parameters $\alpha$ and $\beta$ dictate the range of relative sensing phenomenon. (**a**) Scaled phosphorylation response $[LR^*_2]$ as a function of background ligand concentration $u_0 = [L]_0/K_{d1}$ for different values of $\alpha$ and fold change $u_1/u_0 = 6$ when $\beta$ was fixed to $\beta = 40$. The colors represent scaled phosphorylation response. The dashed green lines sketch the range of concentrations $u_0$ where $[LR^*_2]$ is insensitive to the initial ligand concentration $u_0$. (**b**) Scaled $[LR^*_2]$ as a function of $u_0$ for different values of $\beta$ and fold change $u_1/u_0 = 6$ when $\alpha$ was fixed to $\alpha = 16$. The dashed green lines sketch the range of concentrations $u_0$ where phosphorylation response $[LR^*_2]$ is insensitive to the background ligand levels. The green arrows show the range of background ligand concentrations over which relative sensing holds for the values of $\alpha$ and $\beta$ evaluated from the model fits to the data.

In summary, the analytical model showed that cells adjust the number of receptors on their plasma membranes in response to background ligand exposures through preferential internalization and subsequent degradation of activated receptors. This effectively allows cells to encode the memory of background ligand exposures on the plasma membrane. The model identified two dimensionless aggregate parameters $\alpha$ and $\beta$ that dictate the range of background ligand concentrations over which the signaling network can sense relative changes in extracellular ligand concentration. In agreement with experimental data (**Figure 3** in main text), the model showed that the relative sensing is observed over an order of magnitude in background ligand exposures and is several orders of magnitude below the equilibrium dissociation constant of the ligand with the receptors. Notably, the model showed that relative sensing was robust to variations in $\alpha$ and $\beta$.

## Section 4. An analytical model for receptors internalization-based relative sensing for a case of monomer-activated receptors signaling

Section 4.1 In this section, we show that relative sensing of extracellular ligand concentration can also occur for signaling cascades where receptor phosphorylation (activation) is initiated by ligand-bound receptor monomers instead of dimers. Therefore, receptor dimerization is not a necessary condition for relative sensing. The derivation of analytical expressions for the monomeric case follows the same general logic used in Sections 1 and 2 above.

Similar to the dimer case, we assume that signal transduction to a downstream target P (*Appendix 1—figure 6*) occurs largely through membrane-bound activated receptors and neglect activation due to endosomal phosphorylated receptors.

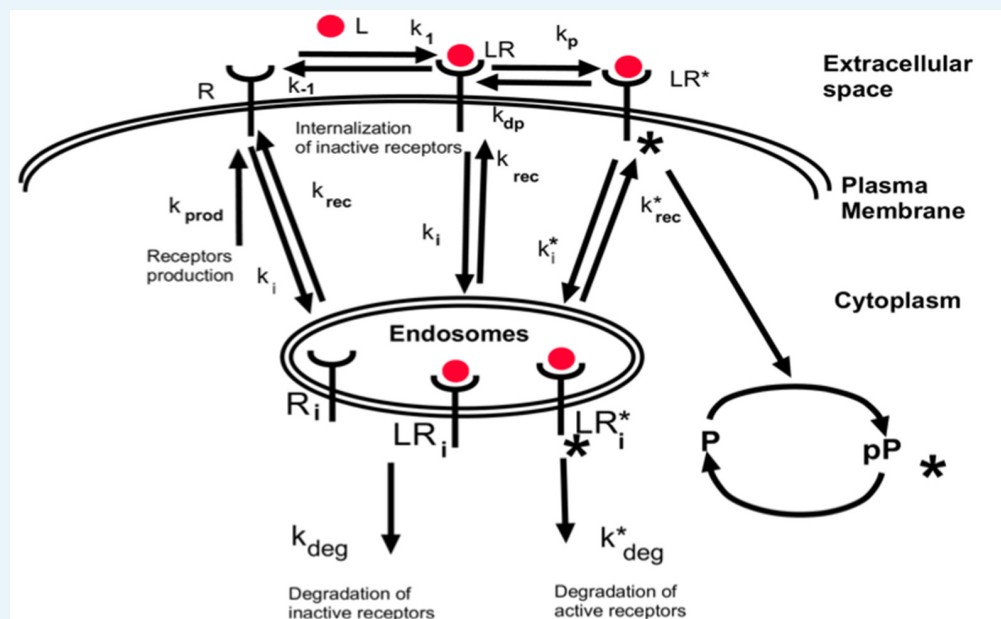

**Appendix 1—figure 6.** Simplified schematics of the steady state analytical model of monomer-activated receptor signaling.

First, we derive the expression for the steady-state cell surface receptor abundance after continuous exposure to the ligand at concentration $[L]_0$. We demonstrate that the receptor level depends approximately logarithmically on the ligand concentration. Then, we derive the expression for the maximum phosphorylation in response to a step increase in extracellular ligand concentration. Notably, we show that the phosphorylation response only weakly depends on the background ligand dose and primarily reflects the fold change in ligand concentration.

The dynamics of species in the model is described by the following equations.

$$\frac{d[R]}{dt} = k_{prod} - k_1[L]_0[R] + k_{-1}[LR] - k_i[R] + k_{rec}[R_i] \tag{A35}$$

$$\frac{d[LR]}{dt} = k_1[L]_0[R] - k_{-1}[LR] - k_i[LR] + k_{rec}[LR_i] - k_p[LR] + k_{dp}[LR^*] \tag{A36}$$

$$\frac{d[LR^*]}{dt} = k_p[LR] - k_{dp}[LR^*] - k_i^*[LR^*] + k_{rec}^*[LR_i^*] \tag{A37}$$

$$\frac{d[R_i]}{dt} = k_i[R] - k_{rec}[R_i] - k_{deg}[R_i] \tag{A38}$$

$$\frac{d[LR_i]}{dt} = k_i[LR] - k_{rec}[LR_i] - k_{deg}[LR_i] - k_p[LR_i] + k_{dp}[LR_i^*] \tag{A39}$$

$$\frac{d[LR_i^*]}{dt} = k_i^*[LR^*] - k_{rec}^*[LR_i^*] - k_{deg}^*[LR_i^*] + k_p[LR_i] - k_{dp}[LR_i^*] \tag{A40}$$

*Equations A35-A40* describe dynamics of [R], ligand-free receptor monomers, [LR], ligand-bound non-activated (unphosphorylated) receptor species, [LR*], ligand-bound activated receptor species, and their internalized counterparts [R_i], [LR_i] and [LR_i*]. The conventions for the rate constants are the same as for the dimer model (section one above).

For convenience, we also introduce the following notations:

$$R_0 = \frac{k_{prod}}{k_i} \times \frac{k_{rec} + k_{deg}}{k_{deg}} \tag{A41}$$

steady-state surface receptor level in the absence of extracellular ligand, $[L]_0 = 0$

$$[R]_T = [R] + [LR] + [LR^*] \tag{A42}$$

the total number of receptors on the cell surface at steady state

$$K_d = \frac{k_{-1}}{k_1} \tag{A43}$$

equilibrium dissociation constant of ligand binding to its receptors

$$u_0 = \frac{[L]_0}{K_d} \tag{A44}$$

the ligand concentration relative to the ligand-receptor dissociation constant

As in the case of dimers-activated signaling, we solved **Equations A35-A40** for the steady state value $[R]_T$ of the cell surface receptor level. Similar to the dimer case, we assume that ligand binding/unbinding and receptor phosphorylation/dephosphorylation happen at a time scale much faster than internalization of inactive receptors. In that limit, we have,

$$[R]_T \approx R_0 \times \frac{1}{1 + \beta \times \kappa \times u_0} \tag{A45}$$

where $\kappa$ and $\beta$ are dimensionless constants given by

$$\beta = \frac{k_i^*}{k_i} \times \frac{\frac{k_{deg}^*}{k_{rec}^* + k_{deg}^*}}{\frac{k_{deg}}{k_{rec} + k_{deg}}}, \qquad \kappa = \frac{k_p}{k_{dp}} \tag{A46}$$

The dependence of $[R]_T/R_0$ on the ligand concentration $u_0$ is shown in **Appendix 1—figure 7**. As an illustration, we use $\beta = 40$ and $\kappa = 20$. Similar to the dimer case, the surface receptor concentration $[R]_T$ depends approximately logarithmically on the ligand concentration near $u_0 \sim 1/\beta\kappa$.

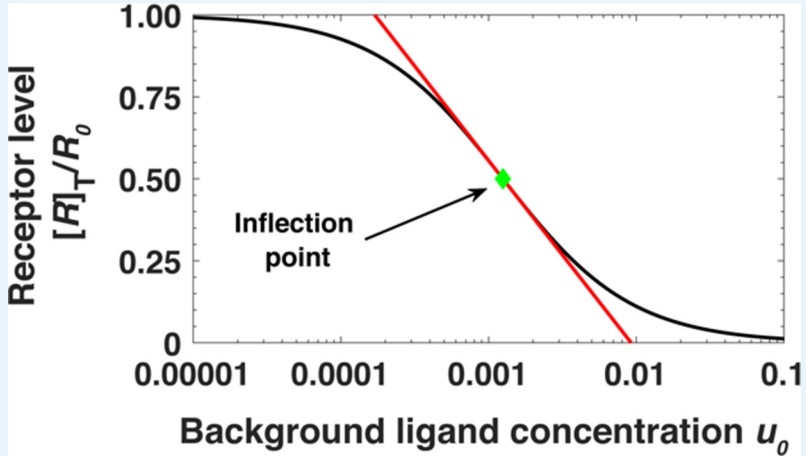

**Appendix 1—figure 7.** Steady state receptor levels decrease proportional to the logarithm of ligand concentration. The steady state cell surface receptor level $[R]_T$ for monomeric receptors (black line) given by **Equation A45** depends approximately on the logarithm of the ligand concentration $u_0 = [L]_0/K_d$ (red line). The green point represents the inflection point.

Section 4.2 Next, using the equation for the total number of cell surface receptors at the steady-state obtained in the previous section we calculated the maximum phosphorylation

level in response to an instant change in ligand concentration from a background level $[L]_0$ to a new level $[L]_1$.

Assumptions. We assume that the dynamic maximum of the phosphorylation response is reached through the rapid equilibration between ligand binding and receptor phosphorylation and dephosphorylation and assume no contribution from relatively slow receptor degradation (**Kleiman et al., 2011**). Consequently, at the time phosphorylated receptors reach their maximum level, the total number of receptors on the cell surface, $[R]_T$, remains approximately the same as the steady state level determined by the pre-exposure to ligand concentration of $[L]_0$. With these approximations, we have at pseudo-equilibrium conditions at the maximum phosphorylated receptor activity:

$$0 = -k_1[L]_1[R] + k_{-1}[LR] \tag{A47}$$

$$0 = k_1[L]_1[R] - k_{-1}[LR] - k_p[LR] + k_{dp}[LR^*] \tag{A48}$$

$$0 = k_p[LR] - k_{dp}[LR^*] \tag{A49}$$

$$[R]_T = [R] + [LR] + [LR^*] \tag{A50}$$

Solving **Equations A47-A50**, we obtain the maximum phosphorylated receptor level

$$[LR^*] = \frac{[R]_T u_1 \kappa}{1 + u_1 + u_1 \kappa} \tag{A51}$$

When the ligand concentration is changed from $u_0 = [L]_0/K_d$ to $u_1 = [L]_1/K_d$ such that $[L]_1/[L]_0 = f$, we have (combining **Equations A45 and A51**)

$$[LR^*] = \frac{u_0 f \kappa}{(1 + u_0 \beta \kappa)(1 + u_0 f(1 + \kappa))} \tag{A52}$$

In **Appendix 1—figure 8**, we plot the maximum phosphorylation response $[LR^*]$ (**Equation A51**) over a range of background concentrations $u_0$ and over multiple folds. We use the parameters obtained for the computational model for EGFR; $\beta = 40$ and $\kappa = 20$ (**Supplementary file 1**).

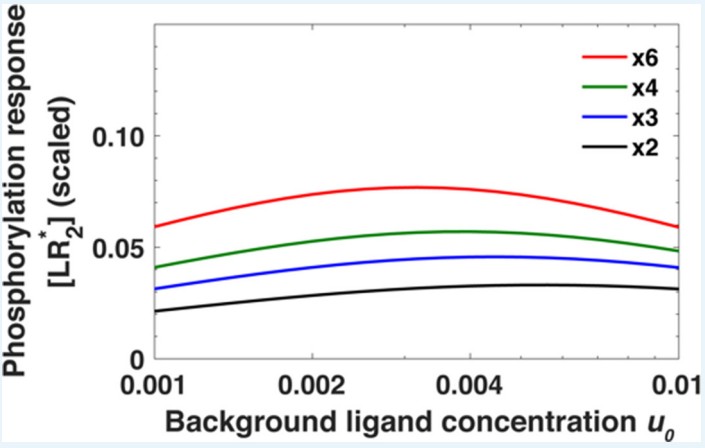

**Appendix 1—figure 8.** Monomeric receptors exhibit relative sensing. The maximum of the phosphorylated receptors level $[LR^*]$ as a function of background ligand concentration $[L]_0$ for fold changes $[L]_1/[L]_0$ in ligand concentration ranging from $[L]_1/[L]_0 = 2$ to $[L]_1/[L]_0 = 6$.

Section 4.3 Similar to the dimeric case, we next explore how changes in the aggregate parameters $\beta$ and $\kappa$ affect relative sensing for monomer-based receptors signaling. In **Appendix 1—figure 8**, we show $[LR^*]$ (scaled) as a function of background ligand concentration at different values of $\kappa$ and $\beta$. As was described in **Appendix 1—figure 5** for dimeric receptors, different colors represent different accuracy values of relative sensing. For

example, the color red represents the range of background ligand concentrations over which the phosphorylation response is within 90% of the maximum (10% deviation). The color yellow represents the range of background ligand concentrations over which phosphorylation response is within 80% of the maximum (20% deviation) and so on. Vertical green lines indicate the region of 90% accuracy; the phosphorylation response $[LR^*_2]$ (scaled) varies within 10% of the maximum. The green arrows show the range of background ligand concentrations over which relative sensing holds for the values of $\alpha$ and $\beta$ evaluated from the model fits to the data.

*Appendix 1—figure 9* panel (a) suggests that the relative sensing range is insensitive to changes in κ when β is fixed. At the same time, when κ is increased the network exhibits relative sensing at lower ligand concentrations. In *Appendix 1—figure 9* panel (b), we show $[LR^*]$ (scaled) as a function of background ligand concentration at different values of β. As β increases, the range of background concentrations over which relative sensing holds increases.

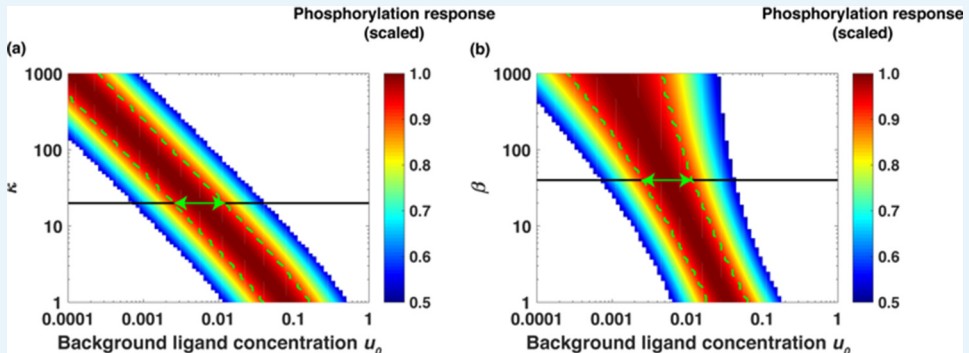

**Appendix 1—figure 9.** Dimensionless parameters κ and β dictate the range of relative sensing. (**a**) Scaled response $[LR^*]$ as a function of $u_0 = [L]_0/K_d$ for different values of κ and fold change $[L]_1/[L]_0 = 2$ when β was fixed at $\beta = 40$. The colors represent scaled phosphorylation response. The green lines sketch the range of concentrations $[L]_0$ where $[LR^*]$ is insensitive to the initial ligand concentration $[L]_0$. (**b**) Scaled $[LR^*]$ as a function of $[L]_0$ for different values of β and fold change $[L]_1/[L]_0 = 2$ when κ was fixed at $\kappa = 20$. The green lines sketch the range of concentrations $[L]_0$ where $[LR^*]$ is insensitive to the initial ligand concentration $[L]_0$. The green arrows show the range of background ligand concentrations over which relative sensing holds for the values of κ and β estimated based on the model fits to the data.

