## [Decision Letter]

**Acceptance summary:**

Accuracy and robustness of biological signaling is an important concept in systems biology that has received significant attention over the years. In this manuscript, the authors present a novel receptor-based mechanism that is sufficient for cells to compute relative changes of growth-factor concentrations in the extracellular milieu (providing approximate fold change detection or FCD). Experimentally, the authors observe increasing pAKT signaling responses that is concomitant with depletion of surface-exposed EGF receptors in cells exposed to increasing concentrations of EGF. Using ODEs coupled with an elegant analytical model and validation experiments, the authors show that surface receptor downregulation is not only a desensitization mechanism, but also a molecular “reference point” as part of a mechanism that compares background concentrations with future stimuli. Receptor-level relative-sensing imbues cells with a sort of molecular memory that can be used to overcome noisy biological conditions independent of transcription.

**Decision letter after peer review:**

Thank you for submitting your article "Receptor-based mechanism of relative sensing and cell memory in mammalian signaling networks" for consideration by *eLife*. Your article has been reviewed by three peer reviewers, and the evaluation has been overseen by a Reviewing Editor and Naama Barkai as the Senior Editor. The following individuals involved in review of your submission have agreed to reveal their identity: Robin E.C. Lee (Reviewer #3).

The reviewers have discussed the reviews with one another and the Reviewing Editor has drafted this decision to help you prepare a revised submission.

Summary:

Accuracy and robustness of biological signaling is an important concept in systems biology that has received significant attention over the years. In this manuscript, the authors present a novel receptor-based mechanism that is sufficient for cells to compute relative changes of growth-factor concentrations in the extracellular milieu (fold change detection of FCD). Experimentally, the authors observe increasing pAKT signaling responses that is concomitant with depletion of surface-exposed EGF receptors in cells exposed to increasing concentrations of EGF. Using ODEs coupled with an elegant analytical model and validation experiments, the authors show that surface receptor downregulation is not a desensitization mechanism, but a molecular “reference point” as part of a mechanism that compares background concentrations with future stimuli. Receptor-level relative-sensing imbues cells with a sort of molecular memory that can be used to overcome noisy biological conditions independent of transcription.

Overall, the work is comprehensive and highlights an important emergent property that arises through receptor endocytosis, a property that may recur in other molecular pathways. Although there is still room for improvements, a suitably revised manuscript would certainly be of interest to a broad biological readership and should be published at *eLife*.

Essential revisions:

1) Review of previous literature:

1a) The Introduction should include a detailed (2-3 paragraph) discussion of fold change detection and its known mechanisms. The present model has approximate (not exact) FCD, and this should be noted.

1b) EGF signaling is one of the better studied signal transduction processes, and multiple observation related to its many facets have been made before. In particular, it has been noted that EGF receptor indeed responds to the EGF dose logarithmically (see e.g., a number of studies from Steve Wiley). These studies are not discussed in the current manuscript, which is quite surprising. The interpretation provided by Wiley and others is that this logarithmic relationship is a consequence of receptors having high and low affinity binding sites, so that the binding of the ligand and the ensuing response depend on a mixture of these binding sites occupied. Needless to say, there is ample literature to support these findings and this model.

We note that a logarithmic dose response is *not* equivalent to FCD (since FCD is a dynamic property), and thus the present work is novel for the EDF system.

2) The analytical modeling is a strong point of this paper, it has a remarkable ability to reproduce experimental findings and explains the range of molecular conditions that support relative sensing. This model should have more page space in the main text. Specifically, the motivation for the analytical modeling can be developed more, the dimensionless parameters α and β can be defined in the main text (they are already summarised indirectly). Figure 7—figure supplement 1 presents important results that can be combined with Figure 4 and discussed in accompanying text.

3) All three reviewers suggested additional experiments as described below. I suggest that the authors add any data they have or can produce in under two months. For experiment where this is not available/possible, I suggest deferring the experiments to future work.

Basically, there is an analysis of wild type cells and an ODE model. The experimental analysis lacks any perturbations to the pathway to really test the ODE model in any particular way. The model analysis lacks distillation to any particular core component or network motif that could be interpreted in a more general manner. One possible experiment to do is overexpression of EGFR, which should keep the cells more sensitive to changes in EGF concentration despite pre-exposure to EGF. An inhibition or knock-down of EGFR should have the opposite effect. Perturbations timed with the fold increase in ligand would have a greater impact. Another experiment would be to pre-expose the cells to EGF for 3 hours, replace with EGF-free media, and measure the rate at which the cell's EGF sensitivity reverts back to baseline levels. According to the authors' model, we should expect to see the EGF sensitivity correlate with the rate at which EGFR is translocated to the cell surface minus the rate at which EGFR is internalized and degraded.

Related to the above, there are multiple pharmacological and genetic methods to perturb receptor trafficking, including its severe inhibition. One needs to test the effects of these perturbations in the overall signaling but also on the specific model predictions, and their validation. The activation of receptor itself upstream of Akt can also be directly tested, e.g., by detecting its phosphorylation status. A better idea of what the pre-stimulation with EGF can do to the cells, including altering the synthesis and degradation rates of the molecules involved in the analysis, and cell behavior (migration, morphology, etc.) should be provided.

The authors emphasize that the proposed relative sensing mechanism is non-transcriptional, but do not provide evidence for this claim. Although the models and experiments demonstrate *sufficiency*, they don't demonstrate *necessity* of a receptor-only mechanism in the axis of EGF/HGF-pAKT-FoxO3 signaling. Note that the timescales mentioned in the Introduction overlap with transcriptional timescales (for example, cytokine-induced transcription can be rapid with strong expression that peaks within 30 minutes – see IL-6 response in PMID: 16191192). I have 2 constructive suggestions: The first would demonstrate necessity by adding additional inhibitor studies: (i) poisoning transcription/translation to demonstrate relative sensing for pAKT/FoxO3 is unaltered; and (ii) inhibiting receptor internalization (MDC, dynasore, etc…) and demonstrating predictable loss of relative sensing (using the model to make predictions). The second suggestion is that the authors can dilute the “non-transcriptional” claims and acknowledge through discussion that transcriptional mechanisms may still supplement the observed non-transcriptional receptor-based mechanism (and explain how future experiments can rule them out).

---

## [Author Response]

Essential revisions:1) Review of previous literature:1a) The Introduction should include a detailed (2-3 paragraph) discussion of fold change detection and its known mechanisms. The present model has approximate (not exact) FCD, and this should be noted.

We have substantially extended our discussion describing previous studies of fold change detection; we made the corresponding changes in the Introduction (paragraph two) and the Discussion section (paragraphs one and two). We have also clarified that our results represent approximate FCD (Results paragraph seven). Notably, we now also show that, in addition to the maximum pAkt response and the integral of pAkt response, the entire time course of pAkt response depends approximately on the fold change of EGF stimulation and not on the absolute EGF levels (Results paragraph seven and Figure 3—figure supplement 6).

1b) EGF signaling is one of the better studied signal transduction processes, and multiple observation related to its many facets have been made before. In particular, it has been noted that EGF receptor indeed responds to the EGF dose logarithmically (see e.g., a number of studies from Steve Wiley). These studies are not discussed in the current manuscript, which is quite surprising. The interpretation provided by Wiley and others is that this logarithmic relationship is a consequence of receptors having high and low affinity binding sites, so that the binding of the ligand and the ensuing response depend on a mixture of these binding sites occupied. Needless to say, there is ample literature to support these findings and this model.

*We note that a logarithmic dose response is* not *equivalent to FCD (since FCD is a dynamic property), and thus the present work is novel for the EDF system.*

We apologize for possible confusion. We have now included several references to previous studies that specifically investigated the biochemical origins of the logarithmic response (Results paragraph two). Importantly, as confirmed by the editor, the logarithmic response is not equivalent and does not guarantee FCD. Thus, previous studies describing the logarithmic response in the system are interesting, but do not compromise the novelty and importance of our work. We have now further clarified this in the manuscript (Discussion paragraph one).

2) The analytical modeling is a strong point of this paper, it has a remarkable ability to reproduce experimental findings and explains the range of molecular conditions that support relative sensing. This model should have more page space in the main text. Specifically, the motivation for the analytical modeling can be developed more, the dimensionless parameters α and β can be defined in the main text (they are already summarised indirectly). Figure 7—figure supplement 1 presents important results that can be combined with Figure 4 and discussed in accompanying text.

We agree with the reviewer. Following the suggestion, we have now included an entire section in the main text, describing in detail the analytical model. We also include the figure illustrating the sensitivity analysis with respect to the two key parameters of the circuit (Results paragraph nine).

3) All three reviewers suggested additional experiments as described below. I suggest that the authors add any data they have or can produce in under two months. For experiment where this is not available/possible, I suggest deferring the experiments to future work.Basically, there is an analysis of wild type cells and an ODE model. The experimental analysis lacks any perturbations to the pathway to really test the ODE model in any particular way. The model analysis lacks distillation to any particular core component or network motif that could be interpreted in a more general manner.

We thank the reviewers for these comments. As we discuss in the main text, the FCD mechanism, based on receptor endocytosis and downregulation, can be effectively described by several equations. The processes of receptors downregulation and receptor-depended signaling are shared across multiple signaling circuits. Therefore, our results are likely to be quite general. We defer further analysis of similar circuits and motifs to our future work.

One possible experiment to do is overexpression of EGFR, which should keep the cells more sensitive to changes in EGF concentration despite pre-exposure to EGF. An inhibition or knock-down of EGFR should have the opposite effect. Perturbations timed with the fold increase in ligand would have a greater impact. Another experiment would be to pre-expose the cells to EGF for 3 hours, replace with EGF-free media, and measure the rate at which the cell's EGF sensitivity reverts back to baseline levels. According to the authors' model, we should expect to see the EGF sensitivity correlate with the rate at which EGFR is translocated to the cell surface minus the rate at which EGFR is internalized and degraded.Related to the above, there are multiple pharmacological and genetic methods to perturb receptor trafficking, including its severe inhibition. One needs to test the effects of these perturbations in the overall signaling but also on the specific model predictions, and their validation. The activation of receptor itself upstream of Akt can also be directly tested, e.g., by detecting its phosphorylation status. A better idea of what the pre-stimulation with EGF can do to the cells, including altering the synthesis and degradation rates of the molecules involved in the analysis, and cell behavior (migration, morphology, etc.) should be provided.

We thank the reviewer for these interesting suggestions. We note that our primary goal in this manuscript was to characterize the behavior of the wild type circuit. To that end, we experimentally measured several key proteins in cells stimulated with two different physiologically important ligands, and multiple fold changes at various background signaling levels. The experiments demonstrated good agreement with our model. We then complemented the experimental measurements with extensive computational modeling, and estimation of multiple systems parameters. Finally, we also performed many detailed analytical derivations.

We agree that perturbing multiple key processes, such as endocytosis, synthesis, and degradation of receptors as well as abundance of other network components are indeed very interesting experiments. However, performing these experiments and their accurate interpretation requires a substantially longer time frame than the under two-month turnaround time specified by the editors. The main reason for this is that perturbations of such key cellular processes is likely to simultaneously change many other components/parameters of the system, such as the relative rates of degradation of active and inactive receptors, the rates of receptor recycling, and potentially the effective phosphorylation and de-phosphorylation rates of various system components. For example, inhibition of clathrin-dependent endocytosis using small molecules inhibitors, such as *dynasore* or *pitstop2*, will potentially affect multiple system processes through a global reorganization of the plasma membrane and the cytosol (see for example [1-4]).

For the wild type network, we relied on many previously measured parameters and their literature estimations, but to properly investigate the effects of endocytosis inhibitors or inhibitors of receptors synthesis we need to measure de novo several key parameters of a perturbed system, or at least validate that these parameters did not substantially change. Experimental measurements of these parameters are long-term projects in themselves, and have been previously published as separate research papers focused on specific parameters, such as receptor degradation and endocytosis rates (see for example, [5-7]). Therefore, we defer detailed analyses of perturbed networks to our future work.

Below we present several important additional experiments that we were able to carefully perform in under two months. One, following the reviewers’ suggestions, confirms a quantitative relationship between the EGFR and Akt phosphorylation. Another, a direct pharmacological activation of Akt, demonstrates that changes at the receptor levels do not affect the inherent activation ability of Akt. Both of these experimental results are essential for our model, but were previously only assumed and not experimentally validated. We also describe experiments, previously performed in our lab on EGFR inhibition, confirming a fast and direct link between EGFR phosphorylation and activation of Akt.

Quantitative relationship between EGFR and Akt phosphorylation:

As the reviewers suggested, it is important to investigate EGFR phosphorylation status, and establish a quantitative relationship between EGFR and Akt phosphorylation. Using quantitative western blot experiments, we have now measured EGFR phosphorylation levels and Akt phosphorylation levels following stimulation with various dosses of EGF (Results paragraph two and Figure 1—figure supplement 3). These experiments demonstrated that Akt phosphorylation levels are approximately linearly related to EGFR phosphorylation levels at the timescales of fast response to EGF stimulation (~5-10 minutes). Notably, this linear relationship was one of the essential components of the model, and was previously assumed to be true without experimental validation.

Direct pharmacological activation of Akt:

We also performed pharmacological perturbation of the system with a small molecule that directly activates Akt regardless of the EGF receptor status. This pharmacological perturbation demonstrated that the desensitization of Akt phosphorylation response, an integral component of the FCD mechanism, was not due to changes in the inherent activation ability of Akt, for example, phosphorylation status-dependent degradation of Akt [8].

We explored direct Akt activation following stimulation with a compound, SC79 [9], which binds to Akt and promotes its activation. Notably, SC79 can activate Akt even in the absence of growth factor stimulation. The direct Akt activation experiments (Results paragraph four and Figure 1—figure supplement 4) demonstrated that while pre-exposure to increasing doses of background EGF desensitizes the pAkt response to further EGF stimulation (decreasing blue bars from left to right in Figure 1—figure supplement 4), SC79 is able to activate Akt to the same extent regardless of the background EGF exposure (similar levels of green bars in Figure 1—figure supplement 4). This confirms another central assumption of our model, i.e. that the desensitization of the circuit does not change the inherent ability of Akt to be activated.

EGFR receptor inhibition:

Previous experiments performed in our lab also validate a direct link between EGF receptor phosphorylation and Akt activation. We previously showed (for the same cell line, MCF10A and the same ligand EGF) that pharmacological inhibition of EGFR phosphorylation, with inhibitors *gefitinib* and *erlotinib*, leads to an almost instantaneous downregulation of EGFR phosphorylation levels (t_1/2_ ~ 10 sec) and a rapid downregulation of Akt phosphorylation levels (t_1/2_ ~ 100 sec) [10].

*The authors emphasize that the proposed relative sensing mechanism is non-transcriptional, but do not provide evidence for this claim. Although the models and experiments demonstrate* sufficiency*, they don't demonstrate* necessity *of a receptor-only mechanism in the axis of EGF/HGF-pAKT-FoxO3 signaling. Note that the timescales mentioned in the Introduction overlap with transcriptional timescales (for example, cytokine-induced transcription can be rapid with strong expression that peaks within 30 minutes – see IL-6 response in PMID: 16191192). I have 2 constructive suggestions: The first would demonstrate necessity by adding additional inhibitor studies: (i) poisoning transcription/translation to demonstrate relative sensing for pAKT/FoxO3 is unaltered; and (ii) inhibiting receptor internalization (MDC, dynasore, etc…) and demonstrating predictable loss of relative sensing (using the model to make predictions). The second suggestion is that the authors can dilute the “non-transcriptional” claims and acknowledge through discussion that transcriptional mechanisms may still supplement the observed non-transcriptional receptor-based mechanism (and explain how future experiments can rule them out).*

We thank the reviewer for the comments and apologize for possible confusion. We agree that cytokine-induced transcription can indeed be fast, although the typical transcriptional response downstream of EGF is likely to take several (6-8) hours [11]. Most importantly, we want to clarify that while the described mechanism is indeed non-transcriptional on the timescales of fast response to an abrupt EGF stimulation (5-15 minutes), transcription and translation play an essential role in the relative sensing circuit that we describe. Specifically, transcription, translation, and delivery of receptors to the cell surface, are all necessary for attaining a background-dependent steady state levels of the membrane receptors, and thus an accurate FCD following further EGF stimulation. We apologize for this confusion and now clarify this in the paper (Discussion paragraph two). As we discussed above, due to the turnaround time frame specified by the editors, we defer experiments on inhibition of transcription and receptor internalization to our future work.

**References**

1) Preta, G., J.G. Cronin, and I.M. Sheldon, Dynasore - not just a dynamin inhibitor. Cell Commun Signal, 2015. 13: p. 24.

2) Basagiannis, D., et al., Dynasore impairs VEGFR2 signalling in an endocytosis-independent manner. Sci Rep, 2017. 7: p. 45035.

3) Ivanov, A.I., Pharmacological inhibition of endocytic pathways: is it specific enough to be useful? Methods Mol Biol, 2008. 440: p. 15-33.

4) Willox, A.K., Y.M. Sahraoui, and S.J. Royle, Non-specificity of Pitstop 2 in clathrin-mediated endocytosis. Biol Open, 2014. 3(5): p. 326-31.

5) Herbst, J.J., et al., Regulation of postendocytic trafficking of the epidermal growth factor receptor through endosomal retention. J. Biol. Chem., 1994. 269(17): p. 12865-73.

6) Lund, K.A., et al., Quantitative analysis of the endocytic system involved in hormone-induced receptor internalization. J Biol Chem, 1990. 265(26): p. 15713-23.

7) Shi, T., et al., Conservation of protein abundance patterns reveals the regulatory architecture of the EGFR-MAPK pathway. Science Signaling, 2016. 9(436): p. rs6.

8) Wu, Y.T., et al., mTOR complex 2 targets Akt for proteasomal degradation via phosphorylation at the hydrophobic motif. Journal of Biological Chemistry, 2011. 286(16): p. 14190-8.

9) Jo, H., et al., Small molecule-induced cytosolic activation of protein kinase Akt rescues ischemia-elicited neuronal death. Proceedings of the National Academy of Sciences, USA 2012. 109(26): p. 10581-6.

10) Kleiman, L.B., et al., Rapid phospho-turnover by receptor tyrosine kinases impacts downstream signaling and drug binding. Mol. Cell, 2011. 43(5): p. 723-37.

11) Brankatschk, B., et al., Regulation of the EGF transcriptional response by endocytic sorting. Sci Signal, 2012. 5(215): p. ra21.